# Neuronal fragile X mental retardation protein activates glial insulin receptor mediated PDF-Tri neuron developmental clearance

Dominic J. Vita [1], Cole J. Meier [1] & Kendal Broadie [1,2,3] ✉

Glia engulf and phagocytose neurons during neural circuit developmental remodeling. Disrupting this pruning process contributes to Fragile X syndrome (FXS), a leading cause of intellectual disability and autism spectrum disorder in mammals. Utilizing a *Drosophila* FXS model central brain circuit, we identify two glial classes responsible for Draper-dependent elimination of developmentally transient PDF-Tri neurons. We find that neuronal Fragile X Mental Retardation Protein (FMRP) drives insulin receptor activation in glia, promotes glial Draper engulfment receptor expression, and negatively regulates membrane-molding ESCRT-III Shrub function during PDF-Tri neuron clearance during neurodevelopment in *Drosophila*. In this context, we demonstrate genetic interactions between FMRP and insulin receptor signaling, FMRP and Draper, and FMRP and Shrub in PDF-Tri neuron elimination. We show that FMRP is required within neurons, not glia, for glial engulfment, indicating FMRP-dependent neuron-to-glia signaling mediates neuronal clearance. We conclude neuronal FMRP drives glial insulin receptor activation to facilitate Draper- and Shrub-dependent neuronal clearance during neurodevelopment in *Drosophila*.

[1] Department of Biological Sciences, Vanderbilt University, Nashville, TN, USA. [2] Kennedy Center for Research on Human Development, Nashville, TN, USA. [3] Vanderbilt Brain Institute, Vanderbilt University Medical Center, Nashville, TN, USA. ✉email: kendal.broadie@vanderbilt.edu

Glial phagocytosis during neural circuit refinement and remodeling occurs from the level of individual synapses through whole neurons[1–3]. A classic example at the peripheral mammalian neuromuscular junction (NMJ) involves glial engulfment to remove excess synapses[4], and similar glial phagocytic activity at the Drosophila NMJ removes shed synaptic material[5]. Centrally, in the developing mammalian retinal geniculate system, glia engulf superfluous axons[6,7], and similar glial phagocytic activity in the developing Drosophila brain mushroom body prunes unneeded axons[8–11]. Importantly, glia can remove entire neurons from neural circuits during developmental remodeling. For example, glia phagocytose neurons in the developing mammalian dorsal root ganglia[12], and similar glial phagocytic activity occurs within the developing Drosophila nervous system[13–17]. It is through such glial elimination that the developing brain sculpts the neural circuitry, facilitating optimized function and behavioral output. Disruptions in glial phagocytosis lead to inefficient connectivity and inappropriate neurons in neural circuits, which are associated with numerous neurological disease states[18–22]. Given the high conservation of glial engulfment processes, Drosophila disease models are emerging as a fruitful avenue to gain genetic and molecular insights into developmental glial phagocytosis mechanisms.

A prevalent neurodevelopmental disorder is Fragile X syndrome (FXS), the most common heritable intellectual disability (ID) and autism spectrum disorder (ASD)[23–25]. Recent reports reveal glial defects across the range of FXS animal models. Fragile X mental retardation protein (FMRP)-knockout mice show defects in glial engulfment of hippocampal synapses[26]. Similarly, Drosophila FMRP nulls display defective glial pruning of mushroom body neurons during metamorphosis, and delayed glial clearance of injured/damaged neurons[27]. Our own work has also suggested putative links to defective glial phagocytic clearance. We discovered that the central brain circuit peptide-dispersing factor tritocerebral (PDF-Tri) neurons, which are normally developmentally transient and eliminated from the circuitry following adult eclosion, fail to prune in Drosophila FMRP mutants and inappropriately persist into maturity[28]. More recently, we discovered that the endosomal sorting complex required for transport III (ESCRT-III)-conserved core component Shrub (human Chmp4) is elevated in the Drosophila FXS model, causing disrupted circuit remodeling with inappropriate axonal branching[29]. Shrub/Chmp4 is known to pinch neuronal membranes to fragment their processes during neuronal clearance[30]. Based on these combined studies, we hypothesized glial phagocytic defects in the Drosophila FXS disease model.

Drosophila has conserved glial phagocytic machinery[1,3,31,32]. For example, the engulfment receptor Draper[33] (mammalian Megf10/Jedi) has an essential conserved role in glial phagocytosis. In Drosophila, Draper mediates glial clearance of central neurons in early development[1,14,34,35], during glial-dependent remodeling of brain circuits during metamorphosis[9,13,17], and also during the elimination of axotomy-damaged neurons following adult eclosion[36–38]. Draper activation initiates a conserved signaling cascade driving the internalization and degradation of phagocytosed material, including Ced-6, Ced-12, Drk, and Src42a proteins[9,39–41]. Proper glial engulfment requires identifying phagocytosis targets via extracellular cues presented by the neurons to glial receptors. Recent Drosophila work has identified insulin-like signaling as a critical cue driving glial phagocytosis following axotomy[42]. Neurons to be phagocytized secrete insulin-like peptides from dense core vesicles, which leads to phosphorylation (activation) of insulin receptors on glia. Insulin signaling drives Draper upregulation in an Akt/Stat92e-dependent mechanism, resulting in glial clearance of neurons[42]. Although insulin signaling has thus been demonstrated for glial responses to brain injury, it remains to be tested whether this mechanism also mediates glial phagocytosis during normal neural circuit developmental remodeling.

In the current study, we focus on developmental remodeling of the Drosophila central brain PDF-Tri neurons[43,44]. These neurons are normally developmentally transient and eliminated over a several-day period following eclosion, but fail to be removed in the Drosophila FXS model[28]. We discover a role for the glial engulfment receptor Draper driving the Dynamin-dependent phagocytic clearance of PDF-Tri neurons during remodeling. We establish a link between FMRP and Draper, with FMRP promoting Draper expression. We show FMRP loss results in reduced Draper levels, while Draper restoration in FMRP-null animals mitigates glial clearance defects in the FXS disease model. We find that FMRP is required only in neurons, not glia, and acts to regulate intercellular signaling that activates glial insulin receptors to initiate the developmental elimination of PDF-Tri neurons. For the phagocytosis mechanism, we show that FMRP-dependent regulation of membrane-molding ESCRT-III Shrub is required, with correction of Shrub levels reducing the inappropriate PDF-Tri neuron retention in the FXS disease model. Taken together, these results reveal a FMRP-dependent neuron-to-glia intercellular signaling pathway triggering glial phagocytic developmental remodeling of central brain neural circuitry, suggesting putative therapeutic targets for the FXS neurodevelopmental disorder.

## Results

**PDF-Tri neurons are eliminated during circuit developmental remodeling.** Development remodeling of brain circuitry requires mechanisms to ensure targeted removal of selected neurons. We focus here on peptide-dispersing factor tritocerebrum (PDF-Tri) neurons, which are pruned following adult eclosion[28,43,44]. To examine these transient PDF-Tri neurons, PDF-Gal4-driven membrane-tethered green fluorescent protein (GFP) (mCD8::GFP) is co-labeled with anti-PDF (Fig. 1)[44,45]. PDF-Tri neurons consist of 1–2 pairs of somata bilaterally positioned in the tritocerebrum on the anterior face of the brain, in close proximity to the esophagus (Fig. 1a)[44]. PDF-Tri processes extend posteriorly and ventrally into the subesophageal zone (SEZ), dorsally through the medial bundle (MDBL), and into the superior medial protocerebrum near the dorsal brain surface (Fig. 1a). The highly ramified dendritic arbors and axonal projections of PDF-Tri neurons project throughout the SEZ just dorsal to, and surrounding, the esophageal foramen (Fig. 1a). As PDF-Tri processes project toward the posterior surface through the MBDL, two linearized branching tracks extend laterally toward the dorsal side of the brain (Fig. 1a). In addition to these developmentally transient PDF-Tri neurons, PDF expression also occurs in the adjacent adult-persistent ventral lateral neurons (LNvs, Fig. 1b). These persistent PDF+ neurons are not the focus of this work, but serve as valuable internal controls for brain labeling and orientation.

Following adult eclosion, the extensive PDF-Tri neuron network is eliminated over a period of several days (Fig. 1b, c)[28,43,44]. In newly eclosed adults (0 days post eclosion, DPE), PDF-Tri neurons are consistently present along the brain central midline (Fig. 1b, c, left). Dense PDF+ projections occur in the SEZ and surrounding the esophageal foramen. Two days later (2 DPE), extensive removal of the PDF-Tri circuitry is underway, with disassembly/process loss following a dorsal-to-ventral gradient (Fig. 1b, c, middle), as reported previously[43]. Higher magnification reveals a fragmented appearance of the mCD8::GFP membrane marker, indicating active remodeling and process elimination during this time window (Fig. 1c, middle). GFP perdurance lasts slightly longer than anti-PDF labeling. Finally by 5 DPE, the PDF-Tri neurons are typically

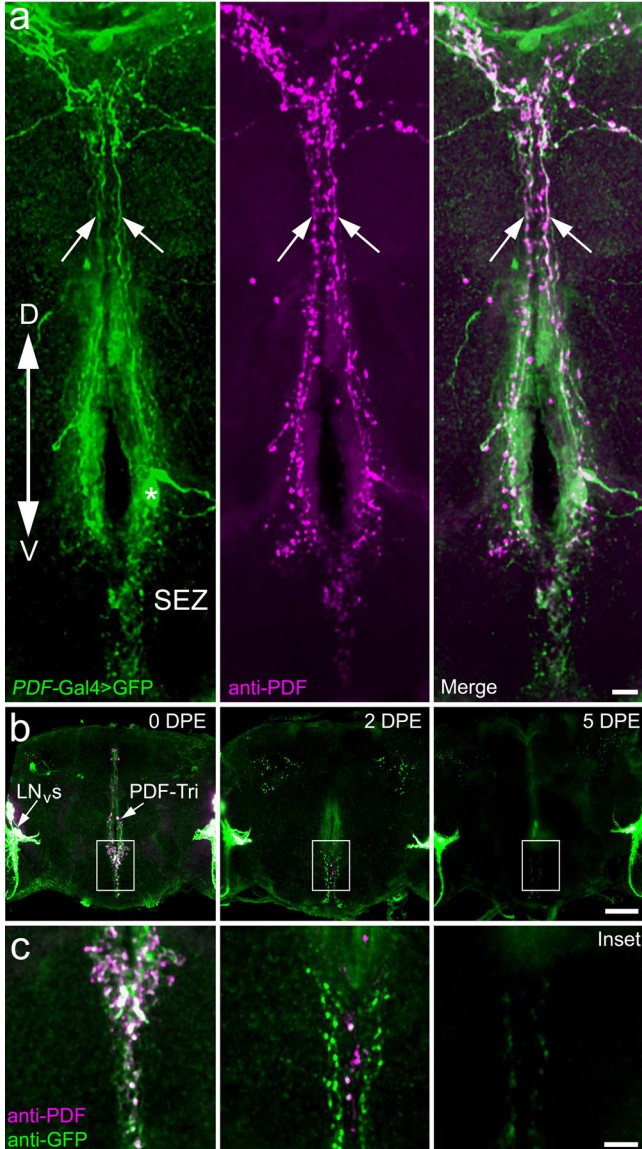

**Fig. 1 Central brain PDF-Tri neurons are transient and eliminated following development. a** Central brain PDF-Tri neurons in a newly eclosed adult *Drosophila* co-labeled for *PDF*-Gal4-driven UAS-mCD8::GFP (*PDF* > GFP, green, left), anti-PDF (magenta, middle), and merged (right). The dorsal (D)–ventral (V) orientation is indicated, with the PDF-Tri neuron soma (asterisk) dorsal to the subesophageal zone (SEZ) and axons projecting via the median bundle (arrows). Scale bar: 10 μm. **b** PDF-Tri neuron clearance shown in days post eclosion (DPE) at 25 °C. Low-magnification whole-brain images are co-labeled for both membrane marker *PDF* > GFP (green) and anti-PDF (magenta), as above in panel **a**, with the developmentally transient central PDF-Tri neurons and adult-persisting ventral lateral neurons (LN$_V$s). Scale bar: 50 μm. **c** High-magnification images of the PDF-Tri neurons (boxed central brain region in panel **b**). The timing sequence shows progressive elimination of the PDF-Tri neurons over the first 5 days of post-eclosion development. Scale bar: 12 μm. Images are representative of three independent experiments.

completely absent from the brain, with only occasional sparse processes remaining (Fig. 1b, c; right). Importantly, both the *PDF*-Gal4-driven mCD8::GFP and anti-PDF labeling of the PDF-Tri neurons have completely disappeared with a similar time course, indicating total elimination of the developmentally transient PDF-Tri neurons by 5 DPE (Fig. 1b, c, right). The other PDF+ clock circuitry persists unaffected. Thus, the PDF-Tri neuron system is

ideally suited for studying the mechanisms of neural circuit developmental remodeling via neuron elimination.

**Draper-dependent glial phagocytosis mediates PDF-Tri neuron elimination.** Glia are the major phagocytes within the brain, recognizing neurons to be phagocytosed by activation of glial engulfment receptors, such as *Drosophila* Draper-I (human Megf10/Jedi)[3]. Upon phagocytosis initiation, glial cells proceed to engulf both neuronal processes and somata via Draper-dependent endocytic mechanisms. The well-studied GTPase "pinchase" Dynamin, first characterized as *Drosophila* Shibire, is essential for the glial phagocytotic engulfment of neurons during this clearance process[8,10,40]. To test whether glia mediate the PDF-Tri neuron elimination, glial phagocytosis was conditionally blocked with a temperature-sensitive *shibire* (*shibire^ts*) allele acting as a dominant negative driven with a pan-glial Gal4 (*repo*-Gal4). Staged animals were raised at the permissive temperature (18 °C) through late pupation, and then shifted to the *shibire^ts* non-permissive temperature (30 °C) just prior to adult eclosion. To test whether Draper is necessary for PDF-Tri neuron elimination, both *draper* global null mutants (*draper^Δ5*) and a characterized *draper*-RNAi targeted only to glia with *repo*-Gal4 were assayed[36]. Staged control and mutant animals were imaged to test for PDF-Tri neuron clearance with anti-PDF labeling at 2 DPE (*shibire^ts*) and 5 DPE (*draper^Δ5* and *draper*-RNAi) when neurons are normally absent. Representative images and quantified results are summarized in Fig. 2.

Glia-targeted *shibire^ts* blockade of phagocytosis causes PDF-Tri neuron clearance defects (Fig. 2a). Compared to controls lacking neurons (left panel), PDF-Tri neurons persist in mutants along the entire central brain midline (right panel, arrows). Like early development stages in controls (Fig. 1), glia-targeted *shibire^ts* results in highest-density PDF-Tri processes within the ventral SEZ (Fig. 2a, lower arrows), compared to dorsal projections (upper arrows). Note that animals were reared at higher temperature (30 °C) to block Dynamin function, and therefore PDF-Tri neuron elimination is accelerated, with controls lacking PDF-Tri neurons by 2 DPE (Fig. 2a). To quantify PDF-Tri neuron clearance, two measurements were made: (1) total PDF-Tri neuron area, and (2) number of PDF+ puncta in the central brain (Fig. 2a, right). In these quantified comparisons, both PDF-Tri area (Mann–Whitney, $p = 0.0006$, $63.52 \pm 17.21$ $n = 18$ control, $412.3 \pm 95.91$ $n = 19$ glial *shibire^ts*) and PDF+ puncta number ($t$ test, $p < 0.0001$, $12.42 \pm 2.012$ $n = 19$ control, $47.15 \pm 6.34$ $n = 20$ glial *shibire^ts*) were significantly elevated in mutant animals compared with the matched controls (Fig. 2a, right). Taking these results together, we conclude that Dynamin-dependent phagocytosis by glial cells that can be conditionally blocked with the temperature-sensitive, dominant-negative *shibire^ts* is required for PDF-Tri neuron elimination.

We next tested Draper glial engulfment receptor involvement in PDF-Tri neuron clearance. In *draper* null mutants (*draper^Δ5*), PDF-Tri neurons are retained at maturity compared to loss in controls (Fig. 2b). Similar to glial *shibire^ts* phagocytosis blockade, PDF-Tri neurons are clearly present in *draper* nulls across the ventral–dorsal axis (Fig. 2b, arrows). PDF+ processes are most prominent in the SEZ and less robustly in the dorsal brain (Fig. 2b). Quantification reveals *draper* nulls have significantly increased PDF-Tri neuron area ($t$ test, $p < 0.0001$, $183.1 \pm 26.78$ $n = 21$ control, $631 \pm 71.56$ $n = 23$ *draper*) and PDF+ puncta (Mann–Whitney, $p < 0.0001$, $28.36 \pm 4.385$ $n = 22$ control, $60.26 \pm 3.661$ $n = 23$ *draper*, Fig. 2b, right). A glial-specific role was tested via *repo*-Gal4-targeted *draper*-RNAi, which causes PDF-Tri neuron clearance defects (Fig. 2c). PDF-Tri processes are prominent in SEZ and surrounding esophageal foramen (arrows).

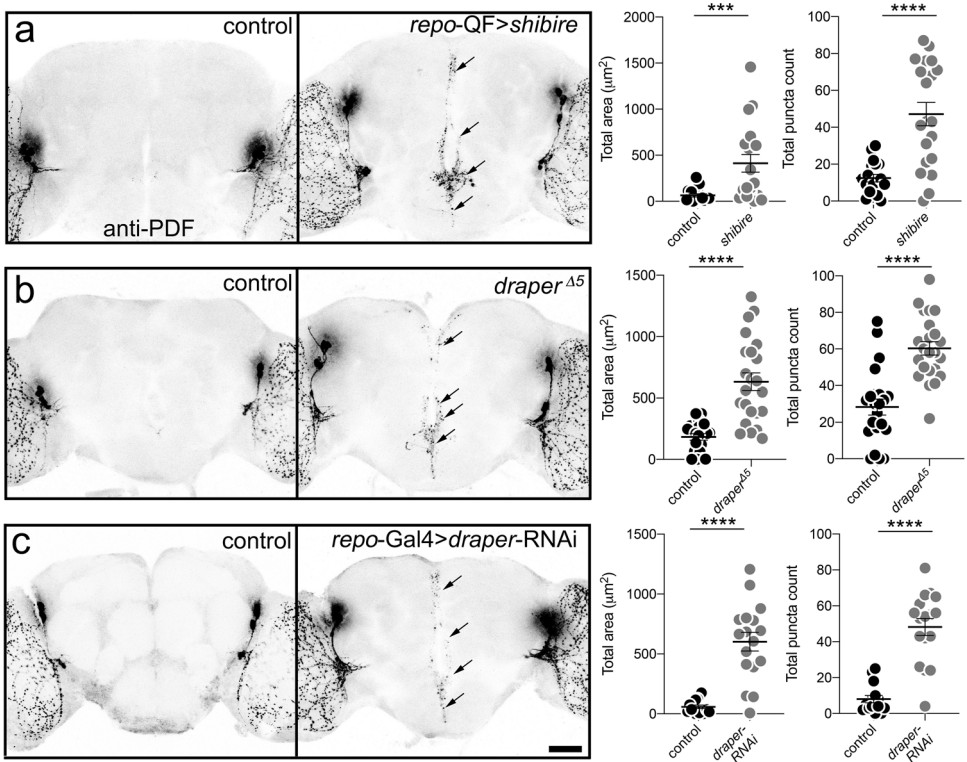

**Fig. 2 Central PDF-Tri neuron elimination requires Draper-dependent glial clearance. a** Whole brains labeled with anti-PDF in transgenic glial driver control (*repo*-QF/+, left) and with dominant-negative *shibire*ts expressed in glia (*repo*-QF > *shibire*ts, right). PDF-Tri neurons (arrows) absent in controls, but persist when glial phagocytosis is blocked. Brains at 2 DPE, raised at 30 °C. Right: Graphs showing central PDF-Tri area (left) and PDF+ puncta (right) in the two genotypes. Area: Two-sided Mann–Whitney, $p = 0.0006$, $63.52 \pm 17.21$ $n = 18$ control, $412.3 \pm 95.91$ $n = 19$ glial *shibire*ts. Puncta: Two-sided *t* test, $p < 0.0001$, $12.42 \pm 2.012$ $n = 19$ control, $47.15 \pm 6.34$ $n = 20$ glial *shibire*ts. **b** Brains labeled with anti-PDF in genetic background control ($w^{1118}$) and *draper* null (*draper*Δ5). PDF-Tri neurons absent in controls, but persist when Draper activity is blocked. Brains at 5 DPE, raised at 25 °C. Right: Graphs show anti-PDF area (right) and PDF+ puncta (left). Area: Two-sided *t* test, $p < 0.0001$, $183.1 \pm 26.78$ $n = 21$ control, $631 \pm 71.56$ $n = 23$ *draper*. Puncta: two-sided Mann–Whitney, $p < 0.0001$, $28.36 \pm 4.385$ $n = 22$ control, $60.26 \pm 3.661$ $n = 23$ *draper*. **c** PDF-labeled brains in transgenic control (*repo*-Gal4/+) and with *draper*-RNAi in glia (*repo*-Gal4 > *draper*-RNAi). Right: Graphs show anti-PDF area (right) and puncta (left). Area: two-sided *t* test, $p < 0.0001$, $58.24 \pm 14.41$ $n = 16$ control, $602.03 \pm 78.46$ $n = 17$ *draper*-RNAi. Puncta: Two-sided *t* test, $p < 0.0001$, $8 \pm 2.04$ $n = 15$ control, $48.24 \pm 4.746$ $n = 17$ *draper*-RNAi. Scatter plot graphs show mean ± SEM. Sample size is $n$ = number of animals. Significance shown for $p < 0.001$ (***) and $p < 0.0001$ (****). Scale bar: 50 μm. Source data for this figure are provided in Source Data file.

The *repo*-Gal4 > *draper*-RNAi appears slightly weaker than the null, likely reflecting incomplete Draper knockdown. Glial-specific Draper knockdown causes significantly more PDF-Tri neuron area (*t* test, $p < 0.0001$, $58.24 \pm 14.41$ $n = 16$ control, $602.03 \pm 78.46$ $n = 17$ *draper*-RNAi) and PDF+ puncta (*t* test, $p < 0.0001$, $8 \pm 2.04$ $n = 15$ control, $48.24 \pm 4.746$ $n = 17$ *draper*-RNAi, Fig. 2c). These results demonstrate that glial Draper-mediated phagocytosis is necessary for PDF-Tri neuron clearance.

We next used terminal deoxynucleotidyl transferase dUTP nick end labeling (TUNEL) to assay for PDF-Tri neuron apoptosis[28] in control versus *draper* null mutants. Brains were labeled at 0 DPE, when PDF-Tri neurons are still present in controls. In both controls and *draper* null animals, the vast majority of PDF-Tri neurons are TUNEL+ (Supplemental Fig. 1a, arrows). Quantification shows no difference in PDF-Tri neuron number (Mann–Whitney, $p = 0.763$, $0.878 \pm 0.0551$ $n = 15$ control, $0.875 \pm 0.0897$ $n = 12$ *draper*) or the percentage of TUNEL+ neurons (Mann–Whitney, $p = 0.227$, $2.333 \pm 0.1260$ $n = 15$ control, $2.000 \pm 0.213$ $n = 12$ *draper*; Supplemental Fig. 1b), indicating normal cell death occurs without Draper function. We examined candidate cell death markers suggested to mark dying cells as Draper ligands. Phosphatidylserine (PS) and Pretaporter are both externalized on apoptotic cells to function as putative Draper ligands[46]. PDF-Tri neurons were labeled for surface

expression of both markers in detergent-free conditions using Annexin-V (PS) and anti-Pretaporter, respectively. There is no detectable co-localization with PDF-Tri neurons (Supplemental Fig. 2a, b). Together, these results show an uncharacterized Draper-dependent mechanism mediates glial phagocytosis of PDF-Tri neurons.

**Ensheathing and cortex glia are the major phagocytes of PDF-Tri neuron clearance.** *Drosophila* has five defined central nervous system glial classes; blood–brain barrier perineurial/subperineural glia, cortex glia (CG) enveloping neural soma, ensheathing glia (EG)-compartmentalizing neuropiles, and astrocyte-like glia (ALG) interacting with synapses[31,47]. We sought to identify the glial class responsible for PDF-Tri neuron clearance. Cortex, ensheathing, and ALG are all implicated as primary phagocytes in various contexts and developmental periods[10,17,34,48]. We therefore focused on these three classes as the candidates. Selective drivers were used for the astrocyte-like (R86E01-Gal4), ensheathing (R56F03-Gal4), and cortex (R54H02-Gal4) glia (Fig. 3a)[47,49]. Combining mCD8::GFP under control of each driver generates distinctive brain localization and cellular morphology for each glial class (Fig. 3a). ALG exhibit a highly ramified and tufted appearance infiltrating central neuropiles (Fig. 3a, left). EG prominently define and surround each brain

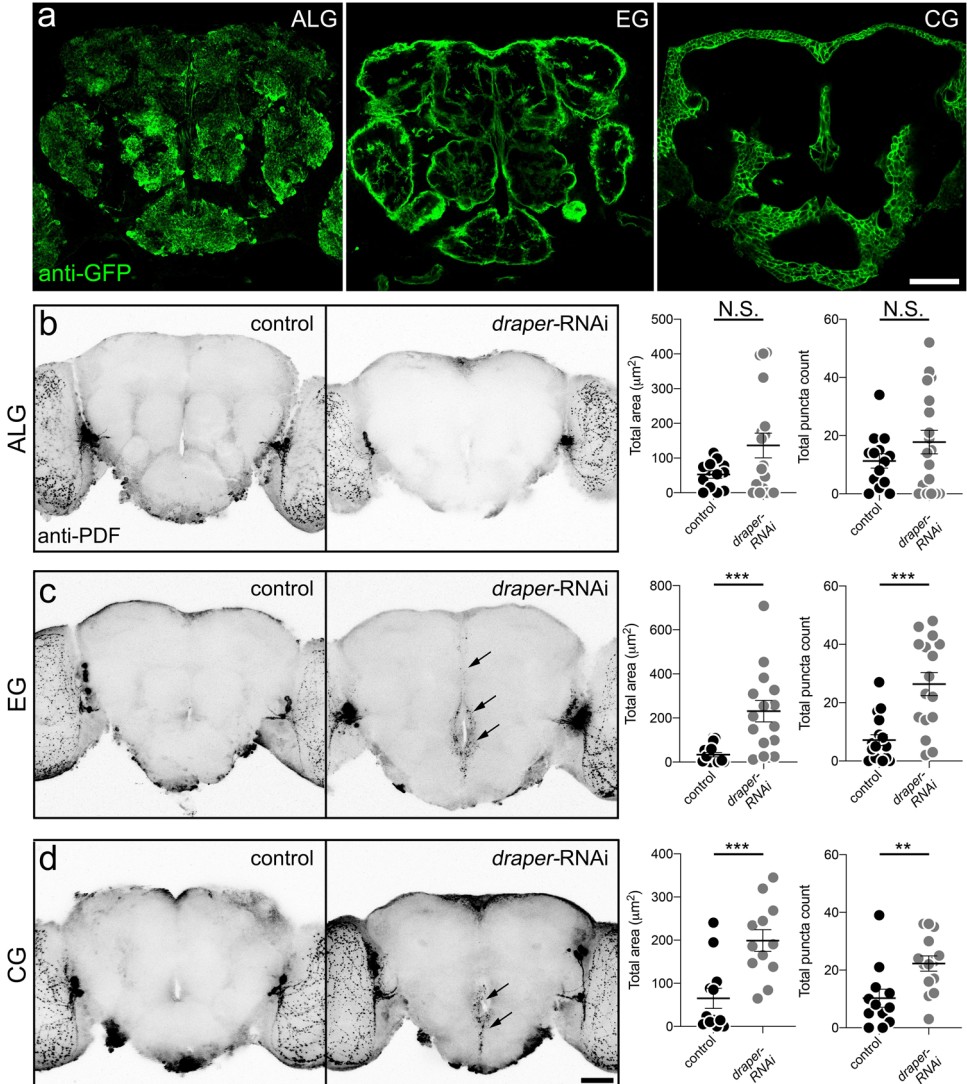

**Fig. 3 Central PDF-Tri neuron elimination requires both ensheathing and cortex glia. a** Brains expressing membrane-tethered mCD8::GFP in glial classes: (1) astrocyte-like glia (ALG, *R86E01*-Gal4 > mCD8::GFP, left), (2) ensheathing glia (EG, *R56F03*-Gal4 > mCD8::GFP, middle), and (3) cortex glia (CG, *R54H02*-Gal4 > mCD8::GFP, right). Images are representative of three independent experiments. **b** 5 DPE brains labeled with anti-PDF in ALG driver control (*R86E01*-Gal4/+) and expressing *draper*-RNAi (*R86E0*-Gal4 > *draper*-RNAi). Right: Graphs show anti-PDF area (left) and puncta (right). Area: Two-sided *t* test, p = 0.0622, 52.55 ± 10.6 *n* = 13 control, 136.3 ± 35.71 *n* = 18 *draper*-RNAi. Puncta: Two-sided *t* test, *p* = 0.214, 11.29 ± 2.471 *n* = 14 control, 17.78 ± 4.068 *n* = 18 *draper*-RNAi. **c** EG driver control (*R56F03*-Gal4/+) and *draper*-RNAi (*R56F03*-Gal4 > *draper*-RNAi). PDF-Tri neurons (arrows) absent in controls, but persist with Draper activity blocked. Right: Graphs show area (left) and puncta (right). Area: Two-sided *t* test, *p* = 0.0003, 33.65 ± 10.14 *n* = 16 control, 230.8 ± 48.787 *n* = 15 *draper*-RNAi; Puncta: Two-sided Mann–Whitney, *p* = 0.0003, 7.176 ± 1.867 *n* = 17 control, 26.38 ± 3.936 *n* = 16 *draper*-RNAi. **d** CG driver control (*R54H02*-Gal4/+) and *draper*-RNAi (*R54H02*-Gal4 > *draper*-RNAi). Right: Graphs show area (left) and puncta (right). Area: Two-sided *t* test, *p* = 0.0008, 65.11 ± 23.33 *n* = 12 control, 199 ± 25.09 *n* = 12 *draper*-RNAi; Puncta: Two-sided Mann–Whitney, *p* = 0.0039, 10.25 ± 3.148 *n* = 12 control, 22.29 ± 2.653 *n* = 14 *draper*-RNAi. Scatter plot graphs: mean ± SEM. Sample size is *n* = number of animals. Significance: *p* > 0.05 (not significant, N.S.), *p* < 0.01 (**), *p* < 0.001 (***), and *p* < 0.0001 (****). Scale bar: 50 μm. Source data for this figure are provided in Source Data file.

neuropile (Fig. 3a, middle). CG are found superficially in the brain surrounding neural cell bodies (Fig. 3a, right). Using each Gal4 driver to target *draper*-RNAi, PDF-Tri neuron clearance defects were assayed with anti-PDF labeling at 5 DPE, when the neurons have been eliminated in controls. Representative images and quantified results are summarized in Fig. 3.

With ALG-driven *draper*-RNAi, PDF-Tri neurons are eliminated normally, similar to controls (Fig. 3b). No statistical difference occurs with ALG knockdown (area: *t* test, *p* = 0.0622, 52.55 ± 10.6 *n* = 13 control, 136.3 ± 35.71 *n* = 18 *draper*-RNAi; puncta: *t* test, *p* = 0.214, 11.29 ± 2.471 *n* = 14 control, 17.78 ± 4.068 *n* = 18 *draper*-RNAi, Fig. 3b, right). In contrast, EG-driven *draper*-RNAi has a strong effect on PDF-Tri neuron clearance

(Fig. 3c). PDF+ processes are easily identified at esophageal foramen and traversing the MBDL (Fig. 3c, arrows). Quantification reveals significantly increased PDF-Tri area and PDF+ puncta (area: *t* test, *p* = 0.0003, 33.65 ± 10.14 *n* = 16 control, 230.8 ± 48.787 *n* = 15 *draper*-RNAi; puncta: Mann–Whitney, *p* = 0.0003, 7.176 ± 1.867 *n* = 17 control, 26.38 ± 3.936 *n* = 16 *draper*-RNAi, Fig. 3c, right). CG-driven *draper*-RNAi also causes PDF-Tri neuron clearance defects. PDF+ processes again occur prominently in the SEZ (Fig. 3d, arrows). CG knockdown results in significantly more PDF-Tri area and PDF+ puncta (area: *t* test, *p* = 0.0008, 65.11 ± 23.33 *n* = 12 control, 199 ± 25.09 *n* = 12 *draper*-RNAi; Puncta: Mann–Whitney, *p* = 0.0039, 10.25 ± 3.148 *n* = 12 control, 22.29 ± 2.653 *n* = 14 *draper*-RNAi, Fig. 3d, right).

These results indicate that both ensheathing and CG phagocytose PDF-Tri neurons.

To test glial class contributions, PDF-Tri neuron clearance was assayed in proximal SEZ and distal MDBL regions (Fig. 4). CG-driven *draper*-RNAi causes retention in SEZ, but not in MDBL (Fig. 4a, left). Quantification shows significantly more PDF area in SEZ compared to MDBL (Mann–Whitney, $p = 0.0006$, $26.61 \pm 17.89$ $n = 7$ MDBL, $94.62 \pm 13.93$ $n = 8$ SEZ) and more PDF+ puncta (Mann–Whitney, $p = 0.0079$, $4.28 \pm 2.643$ $n = 7$ MDBL, $18.88 \pm 3.573$ $n = 8$ SEZ, Fig. 4b). Compared to driver control (*R54H02*-Gal4/+), CG-driven *draper*-RNAi results in significantly more PDF area (Mann–Whitney, $p = 0.0003$, $5.236 \pm 2.481$ $n = 7$ control, $94.62 \pm 13.93$ $n = 8$ *draper*-RNAi) and PDF+ puncta (Mann–Whitney, $p = 0.0020$, $2.429 \pm 1.232$ $n = 7$ control, $18.88 \pm 3.573$ $n = 8$ *draper*-RNAi) only in SEZ (Supplemental Fig. 3). In contrast, EG-driven *draper*-RNAi results in PDF-Tri neuron retention in both SEZ and MDBL (Fig. 4a, right). Compared to driver control (*R56F03*-Gal4/+), EG-driven *draper*-RNAi causes significantly more PDF area (Mann–Whitney, $p = 0.0034$, $13.19 \pm 11.31$ $n = 8$ control, $58.98 \pm 13.89$ $n = 8$ *draper*-RNAi) and puncta (Mann–Whitney, $p = 0.0006$, $1.25 \pm 0.9955$ $n = 8$ control, $13.88 \pm 2.642$ $n = 8$ *draper*-RNAi) in SEZ and MDBL (area: $t$ test, $p = 0.0059$, $6.738 \pm 4.501$ $n = 8$ control, $33.33 \pm 6.844$ $n = 8$ *draper*-RNAi; puncta: Mann–Whitney, $p = 0.00373$, $3.625 \pm 2.464$ $n = 8$ control, $12.00 \pm 2.360$ $n = 8$ *draper*-RNAi) (Supplemental Fig. 3), suggesting a CG/EG glial interaction driving the clearance of PDF-Tri neurons.

We next sought to determine whether the CG or EG class is responsible for removal of PDF-Tri somata. Brains were co-labeled with anti-PDF and anti-Draq5 (nuclear stain) with both CG and EG-driven *draper*-RNAi to assay for PDF-Tri cell body retention (Fig. 4c, d). CG driver controls rarely show PDF/Draq5 colocalization at five DPE (Fig. 4c, top panel). In contrast, CG-driving *draper*-RNAi consistently causes retained PDF-Tri cell bodies (Fig. 4c, lower panel, arrows). In quantified comparisons, CG-driven *draper*-RNAi results in significantly more PDF-Tri neuron somata per brain compared to driver controls (Mann–Whitney, $p = 0.0009$, $0.2308 \pm 0.1662$ $n = 13$ control, $1.643 \pm 0.3249$ $n = 14$ *draper*-RNAi) (Fig. 4c, right). However, examination of EG-driven *draper*-RNAi reveals no detectable effect on PDF-Tri neuron cell body removal, with colocalization between PDF and Draq5 almost never observed in either controls or EG > *draper*-RNAi (Fig. 4d). Quantification demonstrates no significant difference in the number of PDF-Tri neuron somata per brain (Mann–Whitney, $p = 0.1567$, $0.000 \pm 0.000$ $n = 16$ control, $0.1818 \pm 0.1220$ $n = 11$ *draper*-RNAi, Fig. 4d, right). These results suggest the CG and EG glia work cooperatively to completely remove PDF-Tri neurons, with the CG class acting specifically on cell bodies and proximal processes while the EG class facilitates clearance more distally.

**FMRP and Draper interact together to mediate PDF-Tri neuron elimination.** We next set forth to identify triggering mechanisms for Draper-dependent glial engulfment during PDF-Tri neuron phagocytosis. Earlier work demonstrates that FMRP is essential for PDF-Tri neuron elimination[28]. In the FXS disease model lacking FMRP (*dfmr1* null mutants), PDF-Tri neurons are retained at maturity (14 DPE) in multiple null alleles, with the defect fully rescued by reintroduction of wild-type FMRP[28]. We therefore tested whether FMRP and Draper interact in the glial phagocytosis mechanism. To test for interaction, nonallelic complementation testing was performed by combining the two null mutants in *trans*-heterozygous combination (*draper*$^{\Delta 5}$/ *dfmr1*$^{50M}$)[50]. *Trans*-heterozygotes were compared to the two individual heterozygotes, as well as the two homozygous null

mutants. If a clearance defect is observed only in *trans*-heterozygotes, similar to the null mutants but with no effect in the individual heterozygotes, the nonallelic noncomplementation indicates the two gene products interact within the same process[51,52]. As above, we assayed for brain PDF-Tri neuron clearance defects via anti-PDF labeling at 5 DPE, when the neurons have been totally eliminated in the controls. A summary of these analyses in shown in Fig. 5.

Compared to controls (*w*$^{1118}$), null heterozygotes (*draper*$^{\Delta 5/+}$, *dfmr1*$^{50M/+}$) show no defects in PDF-Tri neuron clearance (Fig. 5a, top), with no significant difference (Kruskal–Wallis, $p > 0.9999$; area: $94.75 \pm 25.94$ $n = 25$ control, $25.60 \pm 7.212$ $n = 19$ *draper*$^{\Delta 5/+}$, $66.16 \pm 19.78$ $n = 19$ *dfmr1*$^{50M/+}$; puncta: $15.15 \pm 3.3$ $n = 27$ control, $8.091 \pm 1.931$ $n = 22$ *draper*$^{\Delta 5/+}$, $9.667 \pm 3.067$ $n = 18$ *dfmr1*$^{50M/+}$; Fig. 5b, c). Both homozygous nulls display prominent PDF-Tri neuron clearance defects (Fig. 5a, bottom, arrows), with significantly increased PDF area (Kruskal–Wallis; $p = 0.0068$ $94.75 \pm 25.94$ $n = 25$ control vs. $417.2 \pm 84.86$ $n = 20$ *dfmr1*, $p < 0.0001$ $94.75 \pm 25.94$ $n = 25$ control vs. $553.2 \pm 69.81$ $n = 16$ *draper*, Fig. 5b) and elevated PDF+ puncta (Kruskal–Wallis, $p = 0.176$ $15.15 \pm 3.3$ $n = 27$ control vs. $30.65 \pm 4.944$ $n = 18$ *dfmr1*, $p = 0.0003$ $15.15 \pm 3.3$ $n = 27$ control vs. $45.81 \pm 5.443$ $n = 16$ *draper*; Fig. 5c). *Trans*-heterozygotes (*draper*$^{\Delta 5}$/*dfmr1*$^{50M}$) closely mirror this PDF-Tri neuron retention (Fig. 5a, bottom right, arrows), with both PDF area and PDF+ puncta significantly elevated (area: Kruskal–Wallis, $p = 0.0020$, $94.75 \pm 25.94$ $n = 25$ control, $438 \pm 69.81$ $n = 31$ *dfmr1/draper*; puncta: Kruskal–Wallis, $p = 0.0155$, $15.15 \pm 3.3$ $n = 27$ control, $32.68 \pm 4.069$ $n = 31$ *dfmr1/draper*; Fig. 5b, c). These results indicate a genetic interaction between FMRP and Draper in PDF-Tri neuron clearance.

**FMRP promotes Draper-I expression to mediate PDF-Tri neuron clearance.** FMRP is an mRNA-binding translational regulator[53,54], therefore we next assayed whether FMRP impacts Draper levels. Western blots from control (*w*$^{1118}$) and *dfmr1* null (*dfmr1*$^{50M}$) brain lysates at 0 DPE were analyzed using a well-characterized Draper antibody recognizing all three isoforms (Drpr-I, -II, -III)[55]. In controls, there is a Drpr-I band at the predicted molecular weight (Fig. 6a, top), with a second band representing both Drpr-II and -III (Fig. 6a, bottom), which are too close in size to distinguish, consistent with previous reports[55]. In *dfmr1* null brains, Drpr-I levels are clearly and consistently reduced, while Drpr-II/III appears identical to controls (Fig. 6a, Supplemental Fig. 4). Quantified comparisons normalized to α-Tubulin show a significant decrease in Drpr-I ($t$ test, $p = 0.0001$, $1.000 \pm 0.03750$ $n = 9$ control, $0.7156 \pm 0.04221$ $n = 9$ *dfmr1*), but no detectable change in Drpr-II/III ($t$ test, $p = 0.9076$, $1.000 \pm 0.06112$ $n = 9$ control, $0.9908 \pm 0.04875$ $n = 9$ *dfmr1*; Fig. 6a, right). To complement Draper studies, the same analysis was performed on numerous other components of the glial engulfment pathway (e.g., Ced-12, Ced-6, Src42a, and Drk; Fig. 6b, Supplemental Fig. 5) using characterized antibodies for each protein[9,56–58]. No detectable changes occur for any of these other pathway components in *dfmr1* null brains compared to controls (Fig. 6b). Therefore, in this pathway only Draper-I appears impacted by FMRP loss.

With Draper-I required for glial phagocytosis[55], *dfmr1* null reduced Draper-I levels suggest a mechanistic impairment predicting restored Draper-I should prevent PDF-Tri clearance defects. To test this hypothesis, UAS-*Drpr-I* was targeted to glia with *repo*-Gal4 in the *dfmr1* null. As above, 5 DPE control brains show complete loss of PDF-Tri neurons (Fig. 6c, left) and *dfmr1* nulls retain neurons (Fig. 6c, middle), with significantly more PDF+ area and puncta (ANOVA, $p < 0.0001$; Area: $14.1 \pm 6.406$

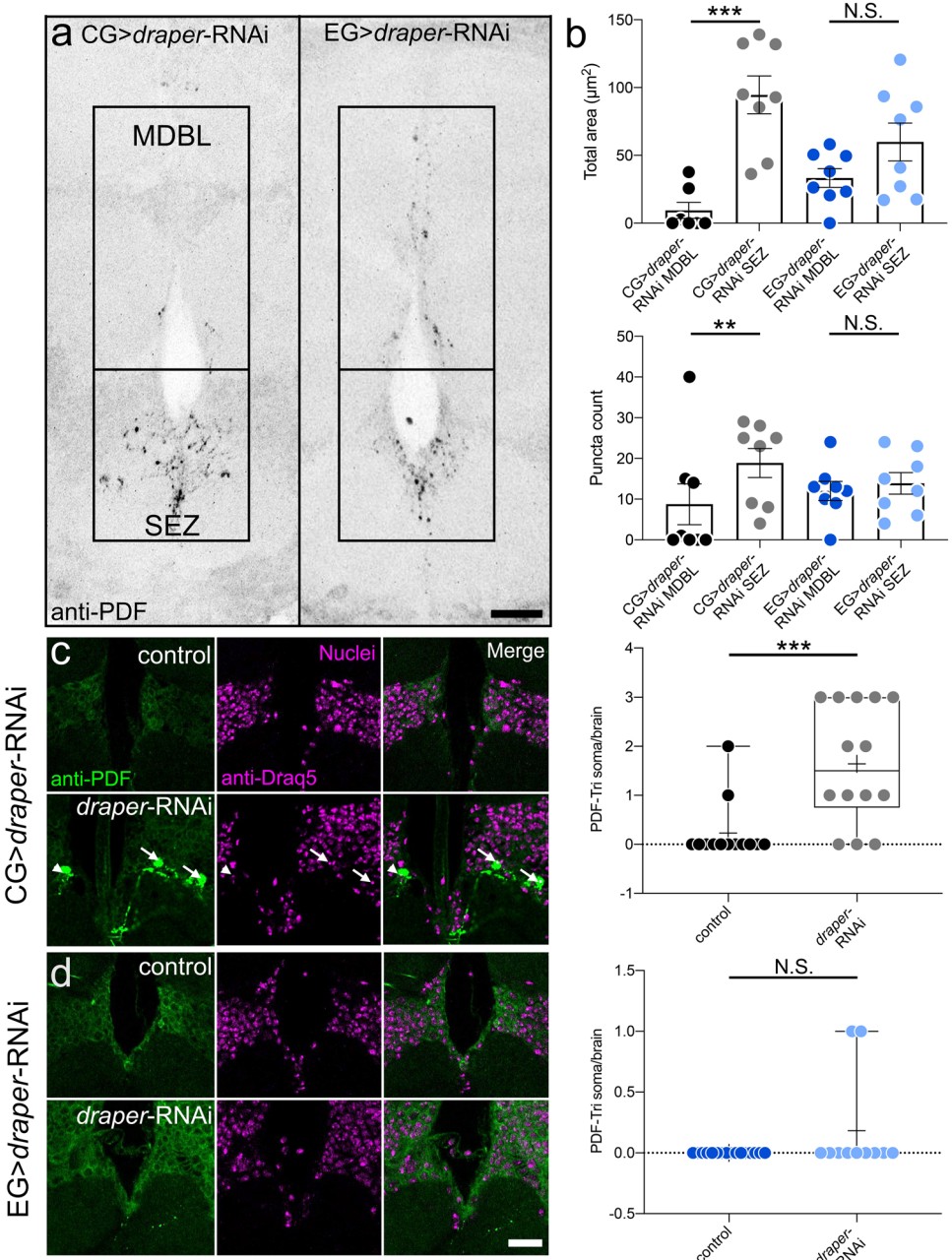

**Fig. 4 Cortex and ensheathing glia clear PDF-Tri neurons in spatially restricted domains. a** Brain midline labeled with anti-PDF at 5 DPE with cortex glia-driven *draper*-RNAi (CG, *R54H02*-Gal4 > *draper*-RNAi, left) or ensheathing glia-driven *draper*-RNAi (EG, *R56F03*-Gal4 > *draper*-RNAi, right). Boxed regions are the distal medial bundle (MDBL) or proximal subesophageal zone (SEZ). Scale bar: 25 μm. Images are representative of two independent experiments. **b** Quantification of anti-PDF area (top) or puncta (bottom) comparing MDBL and SEZ with CG > *draper*-RNAi (left, black/gray) and EG > *draper*-RNAi (right, blue/light blue). Area: CG > *draper*-RNAi, Two-sided Mann–Whitney, $p = 0.0006$, $26.61 \pm 17.89$ $n = 7$ MDBL, $94.62 \pm 13.93$ $n = 8$ SEZ. EG > *draper*-RNAi, Two-sided $t$ test, $p = 0.1084$, $33.33 \pm 6.844$ $n = 8$ MDBL, $59.89 \pm 13.89$ $n = 8$ SEZ. Puncta: CG > *draper*-RNAi, two-sided Mann–Whitney, $p = 0.0079$, $4.28 \pm 2.643$ $n = 7$ MDBL, $18.88 \pm 3.573$ $n = 8$ SEZ. EG > *draper*-RNAi, Two-sided $t$ test, $p = 0.6049$, $12.00 \pm 2.360$ $n = 8$ MDBL, $13.88 \pm 2.642$ $n = 8$ SEZ. **c** Brain slices from 5 DPE control (CG, *R54H02*-Gal4/+, top) and *draper*-RNAi (CG, *R54H02*-Gal4 > *draper*-RNAi) labeled with anti-PDF (green, left), nuclear anti-Draq5 (magenta, middle) and merge (left). Arrows indicate colocalized ant-PDF and -Draq5 (PDF-Tri soma). Arrowhead indicates PDF-Tri neuron slightly out of plane. Right: Quantification of PDF-Tri neuron cell bodies per brain. Two-sided Mann–Whitney, $p = 0.0009$, $0.2308 \pm 0.1662$ $n = 13$ control, $1.643 \pm 0.3249$ $n = 14$ *draper*-RNAi. **d** Brain slices from 5 DPE control (EG, *R56F03*-Gal4/+, top) and *draper*-RNAi (EG, *R56F03*-Gal4 > *draper*-RNAi) labeled with anti-PDF (green, left), nuclear anti-Draq5 (magenta, middle) and merge (left). Scale bar: 25 μm. Right: Quantification of PDF-Tri neuron cell bodies per brain. Two-sided Mann–Whitney, $p = 0.1567$, $0.000 \pm 0.000$ $n = 16$ control, $0.1818 \pm 0.1220$ $n = 11$ *draper*-RNAi. Scatter plots show mean ± SEM. Box and whisker plots show quartiles with max and min values, with plus sign indicating the mean. Sample size is $n$ = number of animals. Significance: $p > 0.05$ (not significant N.S.), $p < 0.01$ (**), and $p < 0.001$ (***). Source data for this figure are provided in Source Data file.

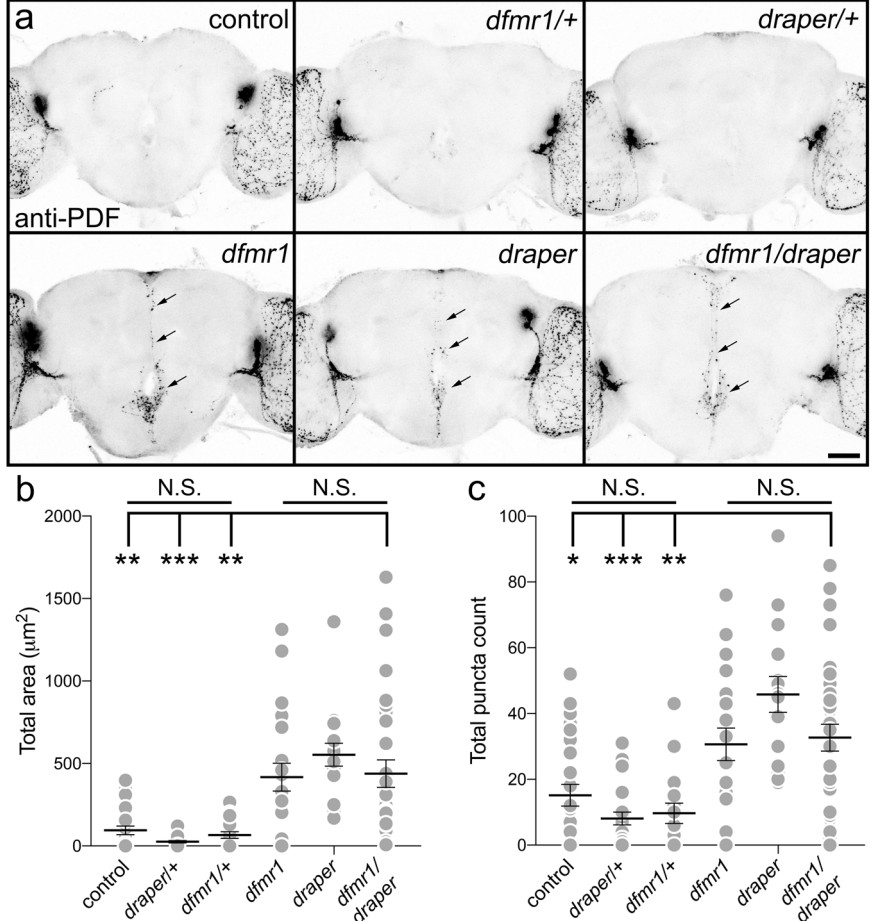

**Fig. 5 FMRP and Draper genetically interact to drive central PDF-Tri neuron elimination. a** Whole brains (5 DPE) labeled with anti-PDF in all controls (top row); genetic background control (w[1118]), dfmr1 null heterozygote control (w[1118]; dfmr1[50M]/+), and draper null heterozygote control (w[1118]; draper[Δ5]/+). All three controls lack the central brain PDF-Tri neurons. The bottom row shows the three experimental genetic conditions; the dfmr1 homozygous null mutant (w[1118]; dfmr1[50M]), the draper homozygous null mutant (w[1118]; draper[Δ5]), and the dfmr1/draper null trans-heterozygote (w[1118]; dfmr1[50M]/draper[Δ5]). All three genotypes exhibit persistent PDF-Tri neurons (arrows). Scale bar: 50 μm. Images are representative of three independent experiments. **b** Quantification of anti-PDF area for the central PDF-Tri neurons in all six genotypes. Kruskal–Wallis followed by Dunn's multiple comparison test, p > 0.9999, 94.75 ± 25.94 n = 25 control, 25.60 ± 7.212 n = 19 draper[Δ5]/+, 66.16 ± 19.78 n = 19 dfmr1[50M]/+, p = 0.0068 94.75 ± 25.94 n = 25 control vs. 417.2 ± 84.86 n = 20 dfmr1, p < 0.0001 94.75 ± 25.94 n = 25 control vs. 553.2 ± 69.81 n = 16 draper, p = 0.0020, 94.75 ± 25.94 n = 25 control, 438 ± 69.81 n = 31 dfmr1/draper. **c** Quantification of anti-PDF puncta number for the central PDF-Tri neurons in all six genotypes. Kruskal–Wallis followed by Dunn's multiple comparison test, p > 0.9999, 15.15 ± 3.3 n = 27 control, 8.091 ± 1.931 n = 22 draper[Δ5]/+, 9.667 ± 3.067 n = 18 dfmr1[50M]/+, p = 0.176 15.15 ± 3.3 n = 27 control vs. 30.65 ± 4.944 n = 18 dfmr1, p = 0.0003 15.15 ± 3.3 n = 27 control vs. 45.81 ± 5.443 n = 16 draper, p = 0.0155, 15.15 ± 3.3 n = 27 control, 32.68 ± 4.069 n = 31 dfmr1/draper. The scatter plot graphs show mean ± SEM. Sample size is n = number of animals. Significance shown for p > 0.05 (not significant, N.S.), p < 0.05 (*), p < 0.01 (**), and p < 0.001 (***). Source data for this figure are provided in Source Data file.

n = 17 control, 560.6 ± 89.03 n = 22 dfmr1; puncta: 3.0 ± 5.347 p = 18 control, 44.95 ± 5.211 n = 22 dfmr1, Fig. 6c, right). In contrast, dfmr1 nulls with glial-targeted Drpr-I exhibit greatly restored PDF-Tri neuron clearance, with only sparse PDF+ process labeling (Fig. 6c, right panel, arrow). Quantification shows glial-targeted Drpr-I in dfmr1 significantly reduces PDF area and PDF+ puncta (area: ANOVA, p = 0.0002, 560.6 ± 89.03 n = 22 dfmr1, 187.4 ± 37.5 n = 17 UAS-Drpr-I; puncta: ANOVA, p = 0.00272, 44.95 ± 5.211 n = 22 dfmr1, 28.67 ± 4.981 n = 21 UAS-Drpr-I; Fig. 6c, right). There is no significant difference in PDF-Tri neuron area compared to driver control (ANOVA, p = 0.152, 14.1 ± 6.406 n = 17 control, 187.4 ± 37.5 n = 17 UAS-Drpr-I), although PDF+ puncta remain elevated (ANOVA, p = 0.0006, 3.0 ± 5.347 n = 18 control, 28.67 ± 4.981 n = 21 UAS-Drpr-I; Fig. 6c, right). These results show loss of glial Draper-I impairs PDF-Tri neuron clearance in the FXS model.

**FMRP acts specifically within neurons for PDF-Tri neuron elimination.** FMRP is classically reported to act in neurons, although more recent accounts demonstrate glial functions[59–61]. To test FMRP requirements in PDF-Tri neuron clearance, dfmr1-RNAi[54] was targeted to glia (repo-Gal4) or neurons (elav-Gal4, Fig. 7). Control and RNAi brains were imaged at 5 DPE, when neurons are normally absent. Driving dfmr1-RNAi in glia has no effect on PDF-Tri neuron clearance, with the neurons absent in repo > dfmr1-RNAi brains (Fig. 7a). Quantification shows no change in PDF area or PDF+ puncta (area: Mann–Whitney, p = 0.5700, 98.62 ± 32.84 n = 15 control, 38.84 ± 16.17 n = 12 dfmr1-RNAi; puncta: t test, p = 0.2650, 14.79 ± 4.432 n = 14 control, 9.063 ± 2.676 n = 16 dfmr1-RNAi, Fig. 7a, right). In sharp contrast, driving dfmr1-RNAi in neurons replicates the dfmr1-null PDF-Tri neuron clearance defect, with PDF-Tri neurons persisting along the entire brain midline (Fig. 7b, arrows).

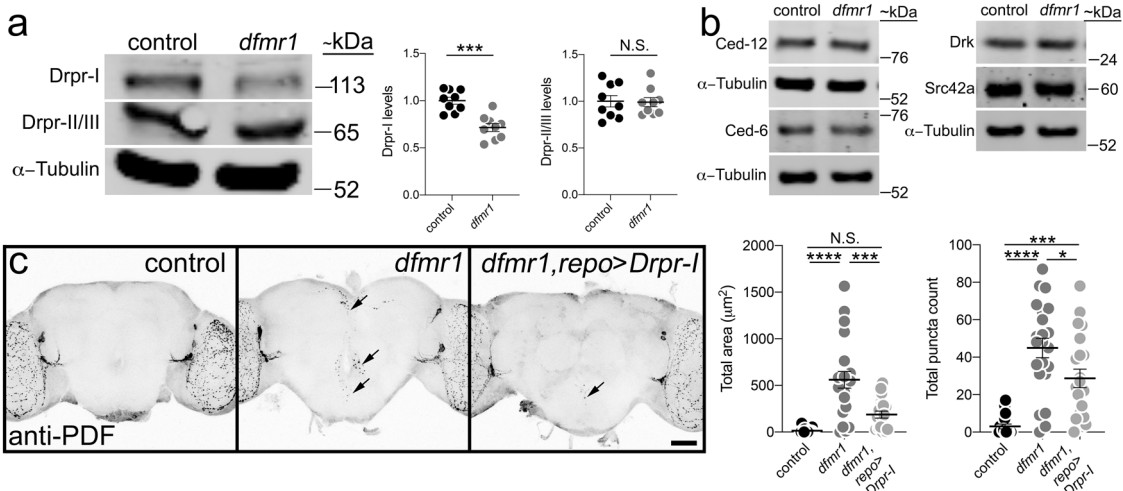

**Fig. 6 FMRP positively regulates Draper-I levels to drive PDF-Tri neuron elimination. a** Representative anti-Draper Western blot of whole-brain lysates (2 brains per lane, 0 DPE) for control (w[1118]) and *dfmr1* null (w[1118]; *dfmr1*[50M]). All bands labeled on the left: Draper-I (Drpr-I, top), Draper-II/III (Drpr-II/III, middle), and α-tubulin (bottom). Molecular weights listed on the right (~kDa). Right: Graphs showing Draper-I (left) and Draper-II/III (right) levels normalized to α-tubulin in both genotypes. Drpr-I: two-sided *t* test, $p = 0.0001$, $1.000 \pm 0.03750$ $n = 9$ control, $0.7156 \pm 0.04221$ $n = 9$ *dfmr1*. Drpr-II/III: two-sided *t* test, $p = 0.9076$, $1.000 \pm 0.06112$ $n = 9$ control, $0.9908 \pm 0.04875$ $n = 9$ *dfmr1*. Sample size is $n =$ number of independent protein extractions (two brains/extraction). **b** Representative Western blots for glial engulfment proteins in control and *dfmr1* null. Labels listed left, and weights on the right. Images are representative of two independent experiments (see Supplemental Fig. 5). **c** Whole brains (5 DPE) labeled with anti-PDF in transgenic glial driver control (*repo*-Gal4/+), *dfmr1* null with glial driver (*dfmr1*[50M], *repo*-Gal4/*dfmr1*[50M]), and driving UAS-*Draper-I* (UAS-*Drpr-I*/+; *dfmr1*[50M], *repo*-Gal4/ *dfmr1*[50M]). Scale bar: 50 μm. Right: Quantification of anti-PDF area (left) and puncta (right). ANOVA followed by Tukey's multiple comparison test. Area: $p < 0.0001$, $14.1 \pm 6.406$ $n = 17$ control, $560.6 \pm 89.03$ $n = 22$ *dfmr1*; $p = 0.152$, $14.1 \pm 6.406$ $n = 17$ control, $187.4 \pm 37.5$ $n = 17$ UAS-*Drpr-I*; $p = 0.0002$, $560.6 \pm 89.03$ $n = 22$ *dfmr1*, $187.4 \pm 37.5$ $n = 17$ UAS-*Drpr-I*. Puncta: $3.0 \pm 5.347$ $p = 18$ control, $44.95 \pm 5.211$ $n = 22$ *dfmr1*; $p = 0.0006$, $3.0 \pm 5.347$ $n = 18$ control, $28.67 \pm 4.981$ $n = 21$ UAS-*Drpr-I*; $p = 0.00272$, $44.95 \pm 5.211$ $n = 22$ *dfmr1*, $28.67 \pm 4.981$ $n = 21$ UAS-*Drpr-I*. Sample size is $n =$ number of animals. Scatter plots show mean ± SEM. Significance: $p > 0.05$ (not significant, N.S.), $p < 0.05$ (*), $p < 0.01$ (**), $p < 0.001$ (***), $p < 0.0001$ (****). Source data for this figure are provided in Source Data file.

Quantification shows PDF-Tri neuron area and PDF+ puncta both significantly increased (area: *t* test, $p = 0.0105$, $108.1 \pm 40.93$ $n = 11$ control, $303.7 \pm 51.3$ $n = 16$ *dfmr1*-RNAi; puncta: *t* test, $p = 0.0018$, $11.91 \pm 3.706$ $n = 11$ control, $36.38 \pm 5.169$ $n = 16$ *dfmr1*-RNAi; Fig. 7b, right). These results show that FMRP is required specifically within neurons, and not in glia, for the developmental elimination of PDF-Tri neurons, indicating that FMRP must regulate neuron-to-glia communication to drive Draper-dependent glial phagocytic clearance.

**FMRP acts via glial insulin receptor activation to mediate PDF-Tri neuron clearance.** Glia identify neurons to be cleared via extracellular cues[3], with insulin receptor (InR) signaling involved in *Drosophila* brain[42]. Compared to 5 DPE controls, InR[E19(HR)] mutants[62] retain PDF-Tri neurons (Fig. 8a). Quantification shows significantly increased PDF area (*t* test, $p = 0.0002$, $39.84 \pm 10.94$ $n = 19$ control, $168.4 \pm 28.86$ $n = 19$ InR[E19(HR)]) and PDF+ puncta (Mann–Whitney, $p < 0.0001$, $6.563 \pm 1.891$ $n = 16$ control, $41.00 \pm 6.841$ $n = 19$ InR[E19(HR)]; Fig. 8b). To test an FMRP/InR signaling intersection, we used a phosphorylated insulin receptor (InR[P]) antibody readout[42]. In controls, prominent InR[P] signaling occurs in *repo*-Gal4-driven mCD8::GFP-marked glial processes (Fig. 9a, top) and a pair of adjacent cell bodies (arrows). To verify these as glia, brains were triple-labeled with *repo*-Gal4-driven mCD8::GFP, anti-Repo, and anti-InR[P]. Colocalization occurs between all three labels (Supplemental Fig. 6a). In *dfmr1* null mutants, activated InR[P] signaling in glia is reduced (Fig. 9a, bottom). Quantification shows InR[P] intensity significantly decreased in *dfmr1* null glia (*t* test, $p = 0.0054$, $1.00 \pm 0.0815$ $n = 12$ control, $0.694 \pm 0.0521$ $n = 11$ *dfmr1*; Fig. 9c). CG- or EG-driven mCD8::GFP glia co-labeled with InR[P] reveal prominent colocalization with EG and sparser

colocalization with CG (Supplemental Fig. 6b, c). Together, these results suggest that InR signaling drives EG/CG glia-dependent PDF-Tri neuron clearance.

We hypothesized that FMRP-dependent neuron-to-glia insulin signaling mediates PDF-Tri neuron clearance, and predicted increasing glia InR activation in *dfmr1* null mutants should prevent phagocytosis defects. To test this idea, a constitutively activated insulin receptor (InRCA)[42] was driven with *repo*-Gal4 in *dfmr1* nulls. 5 DPE controls exhibit PDF-Tri neuron elimination and *dfmr1* nulls retain PDF-Tri neurons (Fig. 9b). Quantification shows *dfmr1* nulls have significantly elevated PDF area (ANOVA, $p = 0.0011$, $18.18 \pm 9.959$ $n = 7$ control, $952.3 \pm 211.9$ $n = 10$ *dfmr1*) and PDF+ puncta (ANOVA, $p < 0.0001$, $3.286 \pm 1.686$ $n = 7$ control, $62.6 \pm 7.524$ $n = 10$ *dfmr1*, Fig. 9d, e). In contrast, glial-targeted InRCA in *dfmr1* nulls restores PDF-Tri neuron clearance, with the neurons mostly eliminated and only weak remnant PDF+ processes persisting (Fig. 9b, right). Quantification shows PDF area significantly reduced compared to *dfmr1* nulls (ANOVA, $p = 0.0197$, $352.8 \pm 105.9$ $n = 10$ InRCA, $952.3 \pm 211.9$ $n = 10$ *dfmr1*), with no significant difference to controls (ANOVA, $p = 0.3183$, $18.18 \pm 9.858$ $n = 7$ control, $352.8 \pm 105.9$ $n = 10$ InRCA, Fig. 9d). PDF+ puncta are also significantly reduced compared to *dfmr1* nulls (ANOVA, $p = 0.0054$, $32.9 \pm 6.28$ $n = 10$ InRCA, $62.6 \pm 7.524$ $n = 10$ *dfmr1*), albeit still slightly elevated compared to controls (ANOVA, $p = 0.0118$, $3.286 \pm 1.686$ $n = 7$ control, $32.9 \pm 6.28$ $n = 10$ InRCA, Fig. 9e). These results suggest neuronal FMRP drives glial InR activation for Draper-dependent phagocytosis of PDF-Tri neurons.

**FMRP regulates ESCRT-III Shrub for PDF-Tri neuron fragmentation and clearance.** FMRP negatively regulates ESCRT-III

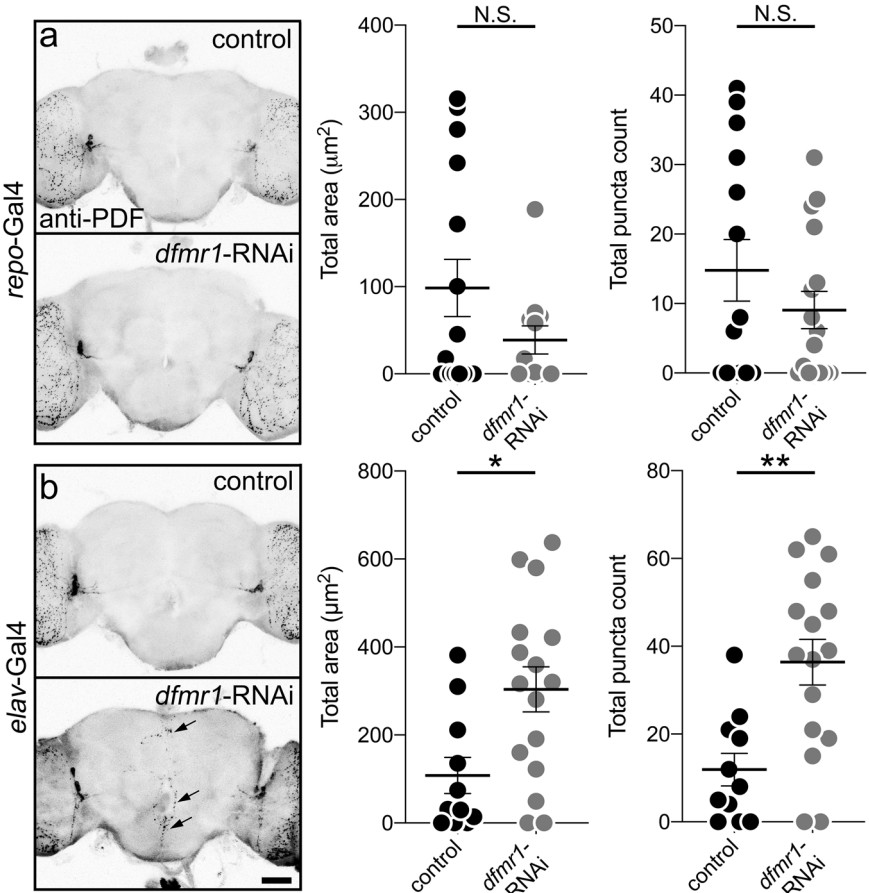

**Fig. 7 FMRP is required specifically within neurons for PDF-Tri neuron elimination. a** Whole brains (5 DPE) labeled with anti-PDF in transgenic glial driver control (*repo*-Gal4/+, top) and expressing *dfmr1*-RNAi targeted to glia (*repo*-Gal4 > *dfmr1*-RNAi[35200], bottom). Both genotypes exhibit a lack of PDF-Tri neurons in the central brain. Right: Graphs showing central PDF-Tri area (left) and PDF+ puncta (right) in the two genotypes. Area: two-sided Mann–Whitney, p = 0.5700, 98.62 ± 32.84 n = 15 control, 38.84 ± 16.17 n = 12 *dfmr1*-RNAi; puncta: two-sided *t* test, p = 0.2650, 14.79 ± 4.432 n = 14 control, 9.063 ± 2.676 n = 16 *dfmr1*-RNAi. **b** Brains (5 DPE) labeled with anti-PDF in the transgenic neuron driver control (*elav*-Gal4/+, top) and expressing *dfmr1*-RNAi targeted to neurons (*elav*-Gal4 > *dfmr1*-RNAi[35200], bottom). The PDF-Tri neurons are absent in controls, but persist when FMRP function is removed in neurons (arrows). Right: Graphs showing anti-PDF area (left) and PDF+ puncta (right) in the two genotypes. Area: two-sided *t* test, p = 0.0105, 108.1 ± 40.93 n = 11 control, 303.7 ± 51.3 n = 16 *dfmr1*-RNAi; puncta: two-sided *t* test, p = 0.0018, 11.91 ± 3.706 n = 11 control, 36.38 ± 5.169 n = 16 *dfmr1*-RNAi. Scatter plot graphs show mean ± SEM. Sample size is n = number of animals. Significance shown for p > 0.05 (not significant, N.S.), p < 0.05 (*), and p < 0.01 (**). Scale bar: 50 µm. Source data for this figure are provided in Source Data file.

core component Shrub/Chmp4[29], which constricts neuronal membranes to drive fragmentation[30]. Importantly, Shrub/Chmp4 loss and overexpression cause similar phenotypes[63,64]. Consistently, Shrub overexpression in PDF-Tri neurons (*PDF*-Gal4 > UAS-*shrub*) results in significant retention (area: *t* test, p = 0.0.0248, 26.85 ± 8.134 n = 14 control, 84.22 ± 18.34 n = 23 UAS-*shrub*; puncta: Mann–Whitney, p = 0.0384, 3.769 ± 0.7775 n = 13 control, 26.39 ± 6.579 n = 23 UAS-*shrub*; Supplemental Fig. 7b, c). Activated Shrub assembles in helical arrays appearing as fluorescent puncta[65,66]. Such Shrub puncta are clearly apparent in PDF-Tri neurons at 2 DPE, indicating Shrub is active during glial developmental clearance (Fig. 10a, top, arrows). High-magnification inspection of all individual Z-stack slices reveals Shrub puncta localized to areas of apparent PDF-Tri neuron fragmentation (Fig. 10a, bottom, arrows). To confirm these observations, we repeated studies with *PDF*-Gal4-driven mCD8:: GFP in PDF-Tri neuron membranes co-labeled with anti-Shrub. Again, Shrub puncta occur at areas of apparent PDF-Tri neuron fragmentation (Supplemental Fig. 7a, arrows). These results suggest a Shrub role in the fragmentation of the PDF-Tri neurons, as a prerequisite for glial engulfment and phagocytic clearance during developmental remodeling.

To test this hypothesis, we reduced Shrub (*shrub*[4]/+)[67] in *dfmr1* null mutants, restoring *dfmr1* Shrub levels to wild-type levels, as previously demonstrated[29]. As above, 5 DPE control animals lack PDF-Tri neurons and *dfmr1* nulls show elevated PDF-Tri neuron area (ANOVA, p = 0.044, 152 ± 35.45 n = 15 control, 632 ± 151.5 n = 16 *dfmr1*) and PDF+ puncta (ANOVA, p = 0.009, 26.88 ± 5.612 n = 16 control, 58.5 ± 8.231 n = 16 *dfmr1*, Fig. 10b). In contrast, Shrub reduction results in strong restoration of PDF-Tri neuron clearance (Fig. 10b, bottom), with loss of PDF-Tri neurons and reduced PDF+ processes (arrow). Quantification shows PDF-Tri neuron area and PDF+ puncta both no longer significantly different from control (area: ANOVA, p = 0.2524, 152 ± 35.45 n = 15 control, 375.6 ± 70.16 n = 18 *shrub*; puncta: ANOVA, p = 0.1698, 26.88 ± 5.612 n = 16 control, 45.06 ± 7.075 n = 18 *shrub*; Fig. 10b), but also not different from *dfmr1* null (area: ANOVA, p = 0.1570, 632 ± 151.5 n = 16 *dfmr1*, 375.6 ± 70.16 n = 18 *shrub*; puncta: ANOVA, p = 0.3720, 58.5 ± 8.231 n = 16 *dfmr1*, 45.06 ± 7.075 n = 18 *shrub*; Fig. 10b). These results indicate that Shrub facilitates PDF-Tri neuron clearance, but may not be absolutely required for glial phagocytosis. Taken together, these findings suggest that ESCRT-III-dependent neural fragmentation facilitates clearance

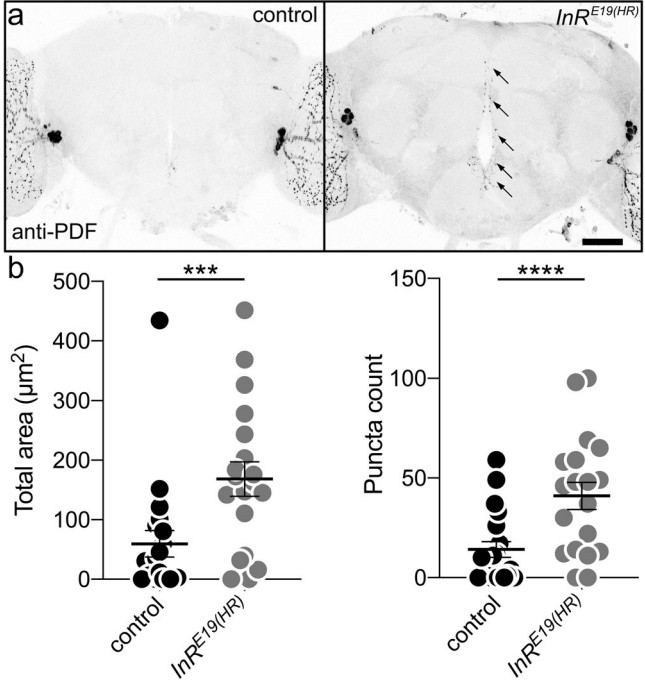

**Fig. 8 Insulin receptor signaling drives developmental clearance of PDF-Tri neurons. a** Whole brains (5 DPE) labeled with anti-PDF in the control (*w*; *InR^wildtype(HR)*, right) and the insulin receptor (InR) mutant (*w*; *InR^E19(HR)*, left). The PDF+ adult-persistent ventral lateral neurons (LN$_v$s) within the lateral brain optic lobes serve as valuable internal controls for both brain labeling and orientation. The developmentally transient PDF-Tri neurons along the central brain midline are absent in the matched genetic background controls, but persist when the InR signaling function is eliminated (arrows). Scale bar: 50 μm. **b** Quantification of anti-PDF area (left) and PDF+ puncta (right) in the two genotypes. Area: two-sided *t* test, *p* = 0.0002, 39.84 ± 10.94 *n* = 19 control, 168.4 ± 28.86 *n* = 19 *InR^E19(HR)*. Puncta: two-sided Mann–Whitney, *p* < 0.0001, 6.563 ± 1.891 *n* = 16 control, 41.00 ± 6.841 *n* = 19 *InR^E19(HR)*. Scatter plot graphs show mean ± SEM. Sample size is *n* = number of animals. Significance: *p* < 0.001 (***) and *p* < 0.0001 (****). Source data for this figure are provided in Source Data file.

of PDF-Tri neurons during developmental remodeling of the PDF circuit.

Axotomy causes insulin-like release for glial InR signaling to drive neuronal clearance[42]. We hypothesized Shrub-dependent fragmentation has a similar role in removal of transient neurons. To test this idea, *shrub^4*/+; *dfmr1* nulls were assayed for glial InR^P activation (Supplemental Fig. 8). Glial InR^P signaling is much reduced in *dfmr1* mutants (Kruskal–Wallis, *p* = 0.0030, 1.00 ± 0.1200 *n* = 8 control, 0.3246 ± 0.04922 *n* = 7 *dfmr1*; Supplemental Fig. 8a, b), but reducing Shrub level mitigates the defect (Supplemental Fig. 8a, bottom), with InR^P levels no longer different from controls (Kruskal–Wallis, *p* = 0.5309, 1.00 ± 0.1200 *n* = 8 control, 0.7064 ± 0.1076 *n* = 9 *shrub*) or *dfmr1* nulls (Kruskal–Wallis, *p* = 0.1138, 0.3246 ± 0.04922 *n* = 7 *dfmr1*, 0.7064 ± 0.1076 *n* = 9 *shrub*; Supplemental Fig. 8b). To confirm this interaction, downstream Draper-I induction was tested in brain lysate Western blots (Supplemental Fig. 9a). Null *dfmr1* mutants have a significant decrease in Draper-I, but no change in Draper-II/III (Draper-I: Kruskal–Wallis, *p* = 0.0160, 1.00 ± 0.1162 *n* = 7 control, 0.8227 ± 0.03697 *n* = 6 *dfmr1*; Supplemental Fig. 9b). Reducing Shrub levels in *dfmr1* nulls block this change, with Draper-I no longer different from controls (Kruskal–Wallis, *p* = 0.6266, 1.00 ± 0.1162 *n* = 7 control, 0.9295 ± 0.03188 *n* = 8 *shrub*; *p* = 0.2870, 0.8227 ± 0.03697 *n* = 6 *dfmr1*, 0.9295 ± 0.03188

*n* = 8 *shrub*; Supplemental Fig. 9b). These findings indicate that FMRP-dependent Shrub regulation modulates the glial InR activation and Draper-I induction driving PDF-Tri neuron clearance.

## Discussion

Developmental remodeling of neural circuitry is a key strategy employed to prune redundant, inappropriate, or interfering neurons in order to optimize connectivity[3]. In the *Drosophila* brain, the PDF-Tri neurons have critical developmental roles, but are eliminated from the adult circuitry[28,43,44]. Clearance occurs rapidly, over a few days, and can be imaged at single-cell resolution (Fig. 1), making this brain circuit a powerful model to dissect remodeling mechanisms. Using conditional blockade of Dynamin function[8,10,40], combined with both global and glia-targeted genetic removal of Draper receptor function[33,34,48], we discover that glial phagocytosis is essential for PDF-Tri neuron elimination (Fig. 2). Of the five known *Drosophila* brain glial classes[31,47], we establish that cortex and EG work cooperatively to mediate PDF-Tri neuron clearance (Figs. 3 and 4). The CG envelop somata[31,47] and EG compartmentalize deep processes[31,47], and we find they act together to remove PDF-Tri neurons extending nearly the entire ventral–dorsal and anterior–posterior brain axes. Similarly, vCrz neurons are phagocytosed by multiple glial classes, with cortex/EG for somata and astrocytes for processes[17]. This work establishes the PDF-Tri neurons as a model for uncovering the mechanism of glia-dependent developmental clearance of neurons in a precisely mapped neural circuit.

Dynamin has well-established roles in glial phagocytosis, acting to constrict inward budding membrane to drive fission[8,10,40]. Dominant-negative *shibire^ts* blockade of Dynamin function has been widely used to test *Drosophila* glial engulfment and phagocytosis roles in early development and following brain injury[8,10,40]. The current study adds Dynamin-dependent glial phagocytosis roles during critical period remodeling in late-stage circuit refinement in a healthy brain. Draper likewise has well-established roles in glial phagocytosis in both mammals and *Drosophila*[3,35,37]. In *Drosophila*, Draper has been extensively studied in the context of neuronal injury, with the receptor binding to cellular debris and/or neural ligands activating a well-characterized signaling cascade leading to Dynamin-dependent internalization[31]. The current study adds to this field by establishing Draper as essential for glial phagocytosis during circuit remodeling in the healthy developing brain. Our results show that glial-specific Draper function is required for the post-eclosion elimination of PDF-Tri neurons, indicating a mechanism exists for these neurons to signal Draper-dependent glial phagocytosis. Disrupting this clearance leads to aberrantly persisting neurons and impaired neural circuit remodeling, defects which have been suggested to contribute to neurodevelopmental disease states like FXS[18,22,23,26].

FXS patients and disease models are characterized by defective neural circuit pruning, with overelaborated axons, dendritic arbors, and supernumerary neurons[18,22,23,26]. In the *Drosophila* FXS model, PDF-Tri neurons are not pruned, but rather maintained into adulthood to remain integrated in the mature circuitry[68]. We find here a link between FMRP loss and Draper loss in the glial phagocytic elimination of the PDF-Tri neurons, with genetic interaction indicating they operate in the same pathway (Fig. 5). Such nonallelic complementation is the best evidence of an in vivo connection, but does not necessitate cell-autonomous interaction. We discover that *dfmr1* mutants have lower Draper-I levels, and that glia-targeted Draper-I expression effectively restores PDF-Tri neuron clearance (Fig. 6). Draper-II/III are not

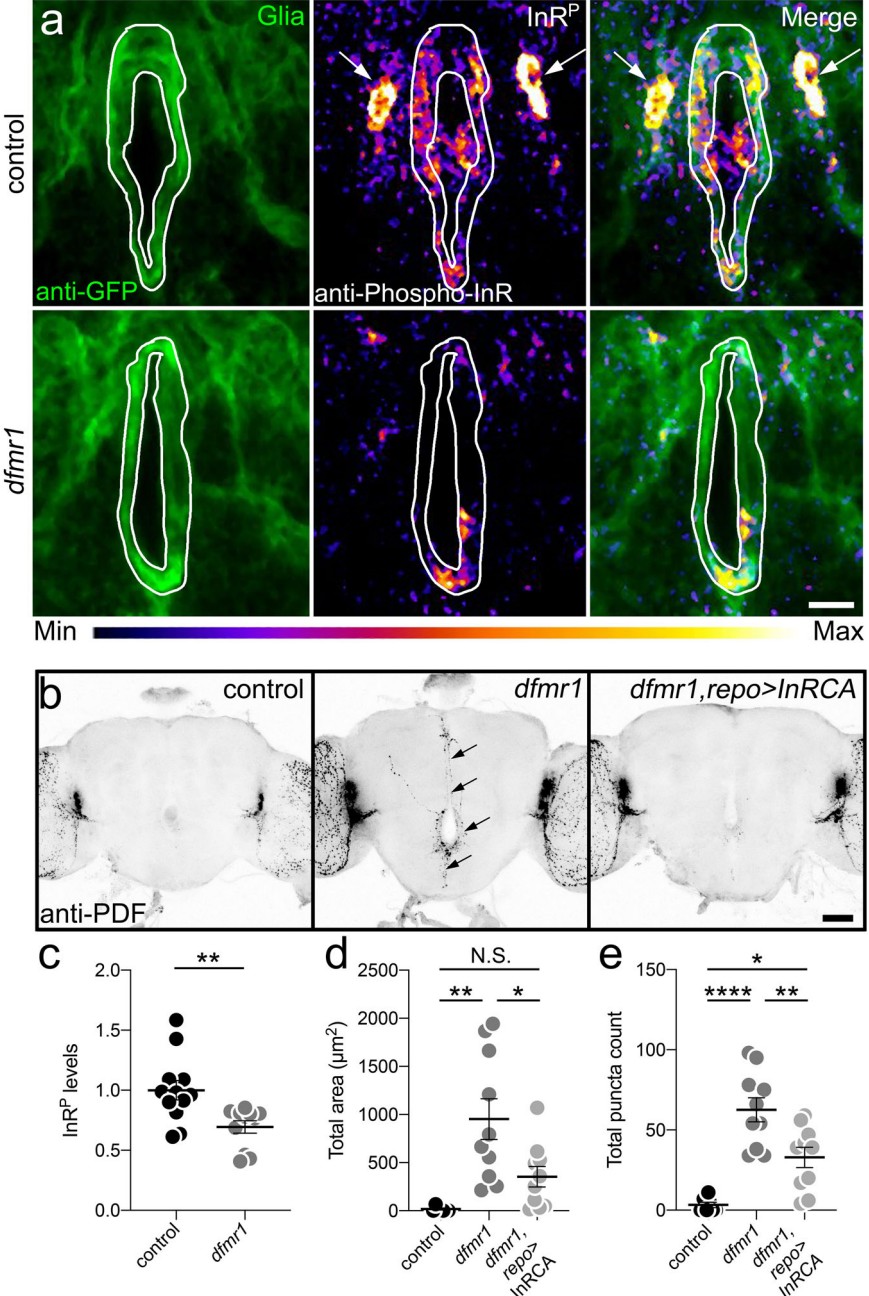

**Fig. 9 FMRP drives glial insulin receptor activation to mediate PDF-Tri neuron clearance. a** Central brains (2 DPE) co-labeled for glial mCD8::GFP (left), anti-phosphorylated insulin receptor (InR$^P$, middle), and merge (right) in the control (UAS-mCD8::GFP/+; repo-Gal4/+) and dfmr1 null (UAS-mCD8::GFP/+; dfmr1$^{50M}$, repo-Gal4/dfmr1$^{50M}$). White outline indicates glial area used for InR$^P$ measurements; arrows indicate glial cell bodies. Range indicator bar for InR$^P$ intensity level below. Scale bar: 10 μm. **b** Whole brains (5 DPE) anti-PDF labeled in glial driver control (repo-Gal4/+), dfmr1 null with driver (dfmr1$^{50M}$, repo-Gal4/dfmr1$^{50M}$), and InRCA targeted to glia (UAS-InR$^{del}$/+; dfmr1$^{50M}$, repo-Gal4/dfmr1$^{50M}$). PDF-Tri neurons (arrows) in dfmr1 null. Scale bar: 50 μm. **c** Quantification of normalized InR$^P$ levels from panel **a**. Two-sided t test, p = 0.0054, 1.00 ± 0.0815 n = 12 control, 0.694 ± 0.0521 n = 11 dfmr1. **d** Quantification of the PDF-Tri area from panel **b**. ANOVA followed by Tukey's multiple comparison test, p = 0.0011, 18.18 ± 9.959 n = 7 control, 952.3 ± 211.9 n = 10 dfmr1; p = 0.3183, 18.18 ± 9.858 n = 7 control, 352.8 ± 105.9 n = 10 InRCA; =0.0197, 352.8 ± 105.9 n = 10 InRCA, 952.3 ± 211.9 n = 10 dfmr1. **e** Quantification of PDF+ puncta from panel **b**. ANOVA followed by Tukey's multiple comparison test, p < 0.0001, 3.286 ± 1.686 n = 7 control, 62.6 ± 7.524 n = 10 dfmr1; p = 0.0118, 3.286 ± 1.686 n = 7 control, 32.9 ± 6.28 n = 10 InRCA; p = 0.0054, 32.9 ± 6.28 n = 10 InRCA, 62.6 ± 7.524 n = 10 dfmr1. Scatter plot graphs show mean ± SEM. Sample sized is n = number of animals. Significance shown for p > 0.05 (not significant, N.S.), p < 0.05 (*), p < 0.01 (**), and p < 0.0001 (****). Source data for this figure are provided in Source Data file.

altered, consistent with Draper-I function in active engulfment, and Draper-II function in phagocytosis cessation[55]. Consistently, the *Drosophila* FXS model also shows reduced Draper-I induction following injury, delaying glial phagocytic response[27]. We find FMRP is required in neurons, not glia, to drive the glial

phagocytosis of PDF-Tri neurons (Fig. 7). Thus, FMRP is not functioning as a *draper* mRNA-binding translational regulator[53], but rather in its capacity as a known regulator of intercellular signaling[69,70]. This discovery reveals a FMRP-dependent neuron-to-glia signaling pathway.

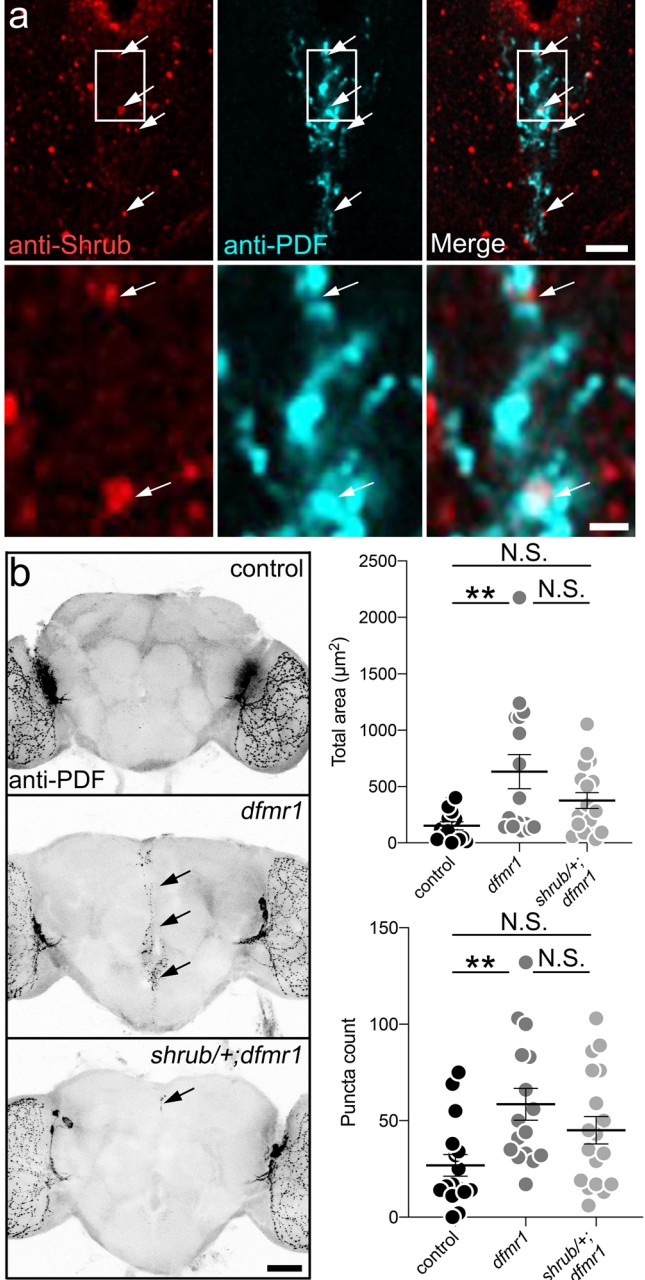

**Fig. 10 FMRP and ESCRT-III Shrub interact to drive central PDF-Tri neuron elimination. a** Brain slices showing the central PDF-Tri region (2 DPE), co-labeled with anti-Shrub (red, left) and anti-PDF (cyan, middle), with merged image (right). Arrows indicate PDF-Tri neuron-localized Shrub puncta. Lower row: higher-magnification images of boxed inset from the upper panels. Scale bars: 10 μm (upper), 2 μm (lower). Images are representative of two independent experiments. **b** Whole brains (5 DPE) labeled with anti-PDF in genetic background control ($w^{1118}$), $dfmr1$ homozygous null mutant ($w^{1118}$; $dfmr1^{50M}$) and heterozygote null $shrub/+$ in the $dfmr1$ homozygous null mutant ($w^{1118}$; $shrub^4/+$; $dfmr1^{50M}$). PDF-Tri neurons absent in control and persistent in $dfmr1$ nulls (arrows) are partially eliminated when one copy of the ESCRT-III core component Shrub is removed (bottom). Scale bar 50 μm. Right: Graphs show the anti-PDF area (top) and puncta (bottom) for all three genotypes. Area: ANOVA followed by Tukey's multiple comparison test, $p = 0.044$, $152 ± 35.45$ $n = 15$ control, $632 ± 151.5$ $n = 16$ $dfmr1$; $p = 0.2524$, $152 ± 35.45$ $n = 15$ control, $375.6 ± 70.16$ $n = 18$ $shrub$; $p = 0.1570$, $632 ± 151.5$ $n = 16$ $dfmr1$, $375.6 ± 70.16$ $n = 18$ $shrub$. Puncta: ANOVA followed by Tukey's multiple comparison test, $p = 0.009$, $26.88 ± 5.612$ $n = 16$ control, $58.5 ± 8.231$ $n = 16$ $dfmr1$; $p = 0.1698$, $26.88 ± 5.612$ $n = 16$ control, $45.06 ± 7.075$ $n = 18$ $shrub$; $p = 0.3720$, $58.5 ± 8.231$ n $= 16$ $dfmr1$, $45.06 ± 7.075$ $n = 18$ $shrub$. The scatter plots show mean ± SEM. Sample size is $n =$ number of animals. Significance shown for $p > 0.05$ (not significant, N.S.) and $p < 0.01$ (**). Source data for this figure are provided in Source Data file.

healthy brain as well. Our results appear at odds with previous work indicating an elevation of insulin-like signaling in the *Drosophila* FXS model brain[72]. However, the previous study focused only on maturity, whereas we show a downregulation in glial InR signaling during developmental remodeling.

FMRP impacts insulin-like signaling in multiple disease models. FMRP KO mice exhibit less insulin growth factor 2 (IGF2) in the hippocampus, with IGF2 treatment improving learning/memory deficits[71]. Similar results in other mouse autism models indicate IGF2 ameliorates a number of cognitive and social interaction impairments[73]. *Drosophila* has 8 Ilps, with some Ilps (e.g., Ilp6) reported to be IGF-like with activities similar to mammalian IGF[74]. FMRP likely controls Ilp release via dense core vesicle (DCV) exocytosis in neurons. During *Drosophila* axotomy experiments, severed axons release IIp-containing DCVs, which in turn interact with glial insulin receptors to promote the phagocytic clearance of damaged neurons[42]. *Drosophila* lacking FMRP display a delayed Draper glial response to injury[27], and FMRP-dependent Ilp release from neurons likely acts upstream of this clearance mechanism. Consistently, a recent report showed impaired DCV release in FXS model mice, based on stimulated synaptoneurosomes from FMRP KO mice exhibiting reduced peptide secretion[75]. Given FMRP null neurons exhibit reduced DCV release of neuropeptides, we suggest FMRP similarly facilitates Ilp-containing DCV release. Future work will test this intriguing mechanism, and identify the specific llps directing glial phagocytosis of PDF-Tri neurons during brain circuit developmental remodeling.

FMRP acts as a negative translationally regulator of conserved ESCRT-III core component Shrub/Chmp4 during *Drosophila* brain circuit remodeling[29]. Shrub has characterized roles in circuit pruning, acting locally on neural branches to drive process fragmentation prior to glial clearance[30]. We show here that Shrub is present during PDF-Tri neuron clearance, appearing in concentrated puncta at neuronal fragmentation sites (Fig. 10). Activated Shrub monomers organize into helical arrays that deform the plasma membrane, constricting processes to fragment neurons[30,66]. Consistent with several previous studies[63,64], Shrub overexpression phenocopies loss-of-function to impair PDF-Tri

Neuron-to-glia insulin receptor (InR) signaling promotes glial phagocytosis of neurons, acting via Stat92e to elevate Draper levels during axotomy[42]. We show here that InR signaling mediates developmental pruning, suggesting a broad role for InR signaling in neuronal clearance. Neurons secrete insulin-like peptides (Ilp) that activate glial InRs to induce Draper-1[42]. We discover here that InR signaling drives PDF-Tri neuron elimination, with glial InRs similarly activated during the critical developmental window (Figs. 8 and 9). We find glial InR activation reduced in the FXS model, consistent with known FMRP roles in insulin signaling[71]. We show neuronal FMRP drives glial InR activation to signal phagocytic clearance of PDF-Tri neurons, and that activating glial InR signaling in the FXS model restores neuronal clearance (Fig. 9). These findings are consistent with previous work showing InR phosphorylation triggers glial phagocytosis[42]. This work shows a role for glial InR activation only in the clearance of injury-damaged neurons, whereas we find glial InR signaling necessary for normal circuit remodeling within the

neuron developmental clearance. We find genetic reduction of Shrub in the FXS model has the positive effect of reducing PDF-Tri neuron clearance defects (Fig. 10). Our results indicate Shrub likely acts to fragment PDF-Tri neurons to facilitate glial phagocytosis. In addition, Shrub also acts to stimulate InR signaling and Draper-I induction in glia, similar to the stimulated Ilp release during axotomy studies[42]. In conclusion, we present a pathway of developmental remodeling of brain circuitry, in which neuronal FMRP drives neuron-to-glial insulin signaling for Draper- and Shrub-dependent glial phagocytosis of developmentally transient neurons.

## Methods

**Drosophila genetics.** Animals were housed at 25 °C with a 12-h light:dark cycle on standard cornmeal, agar, and molasses food media. For conditional studies with temperature-sensitive (ts) shibire[ts] mutants[8,10,40], animals were raised at 18 °C with 12-h light:dark cycle until pharate pupal day 4 (D4), then transferred to 30 °C for 2 days before analyses. All lines were generated from the following stocks via either recombination or genetic crossing: (1) w[1118] [68]; (2) y,w;pin[1]/cyo;repo-QF2/Tm6,tb (BDSC#66477); (3) QUAS-shibire[ts] (BDSC#30012); (4) wg,sp/ cyo,draper[Δ5]/Tm6,Sb, Tb[36]; (5) repo-Gal4/Tm3,Sb (BDSC#7415); (6) y,w; UAS-draper-RNAi/cyo[36]; (7) w[1118]; R86E01-Gal4 (BDSC#45914); (8) w[1118]; R56F03-Gal4; (BDSC#39157); (9) w[1118]; R54H02-Gal4 (BDSC#45784); (10) w[1118]; dfmr1[50M]/Tm6,Tb,GFP[50]; (11) y,w; UAS-Draper-I (BDSC#67035); (12) w[1118]; PDF-Gal4[28]; (13) w[1118]; UAS-mCD8::GFP[45]; (14) w[1118]; elav-Gal4[29]; (15) y,w; UAS-dfmr1-RNAi (BDSC#35200); (16) y,w; UAS-InR[Del] (BDSC#8248); (17) w[1118]; shrub[4]/cyo[67]; (18) w[1118]; UAS-shrub[67]; (19) w[1118]; InR[E19(HR)] 62; and (20) w[1118]; InR[WildType(HR)] 62.

**Confocal imaging.** Immunohistochemistry was performed as follows. Brains from staged animals were dissected in phosphate-buffered saline (PBS), then fixed in 4% paraformaldehyde (PFA) + 4% sucrose in PBS (pH 7.4) for 30 min at room temperature (RT) with end-over-end rotation. The preparations were then washed 3× in PBST (0.2% Triton-X 100 in PBS), followed by 2 h blocking (1% bovine serum albumin (BSA) + 0.5% normal goat serum (NGS) in PBST) at RT, with rotation as above. Preparations were then incubated overnight with primary antibodies diluted in 0.2% BSA, 0.1% NGS in PBST at 4 °C, with rotation as above. Preparations were washed 3× for 20 min at RT with rotation in PBST, then incubated in secondary antibodies for 4 h at RT, with rotation as above. Preparations were then washed as above with an additional final wash in PBS for 20 min. Preparations were mounted in Fluoromount (EMS 17984-25) under a glass coverslip (No. 1.5H, Carl Zeiss), with double-sided tape to act as a spacer between brain and coverslip. Slides were sealed with clear nail polish. For nuclear labeling (Fig. 4c, d), brains were incubated in Draq5 (Abcam, AB108410, 1:250) in PBS for 30 min at RT after the final PBS wash above, and then mounted in 75% glycerol in PBS. Slides were imaged using a510 META laser-scanning confocal microscope (Carl Zeiss) with 20× air or 40× oil-immersion objectives. All images were collected at 1072 × 1072 resolution with optical slice thickness set via optimization function on Zen software. Image parameters were kept constant within replicates. Primary antibodies used: rabbit anti-GFP (Abcam290, 1:1000), chicken anti-GFP (Abcam, AB13970), mouse anti-Repo (Developmental Studies Hybridoma Bank (DSHB), 8D12, 1:500), mouse anti-PDF (DSHB, PDF C7, 1:5), mouse anti-Draper (DSHB, Draper 8A1, 1:500), rabbit anti-phospho-InR β (Cell Signaling, Tyr1146, 1:1000), and rabbit anti-Shrub (1:500; a generous gift from Fen-Bioa Gao, UMass Medical, USA)[67]. Secondary antibodies used: AlexaFlour 488 goat anti-rabbit, AlexaFlour 488 goat anti-mouse, AlexaFlour 568 goat anti-mouse, AlexaFlour 633 goat anti-mouse, AlexaFlour 568 goat anti-rabbit, and AlexaFlour 488 goat anti-chicken. All secondaries were used at 1:250. For PDF-Tri neuron analyses, ImageJ software was used to generate max intensity projections from ten slices entirely encompassing the neurons. Images were processed 3× with the despeckle tool, cropped to identical height/width, binarized via threshold adjustment, and then measured using the particle analyzer plugin. For InR[P] analyses, region of interests were selected using the ImageJ freehand tool to trace labeled glia at the subesophageal foramen. Intensity measurements were made from ten slices in a Z-stack, then averaged for each brain. Measurements were normalized to controls for cross-comparisons between trials. All image analyses were performed blind.

**TUNEL studies.** Terminal TUNEL was used to assay for apoptosis[76]. Experiments were carried out according to the manufacturer's instructions (Roche, 12156792910), with slight modifications. Staged brains were prepared through secondary antibody labeling as above. Brains were then incubated at a 1:100 mix with 10% Triton-X 100 and sodium citrate at 65 °C for 30 min. Brains were then washed 3× with PBS, followed by incubation with 45 μl of labeling solution for 30 min at 35 °C. In all, 5 μl of the enzyme was then added, and preparations were allowed to incubate for 2 h at 35 °C. Following incubation, brains were washed 3× in PBS, then mounted in Fluoromount, and imaged as above.

**Annexin V and pretaporter.** Annexin V was used to assay for externalized PS[46]. Brains were processed through fixation as above, then labeled with Annexin V according to the manufacturer's instructions (BD Pharmingen, 556547). Brains were then secondarily fixed with 4% PFA in the Annexin V binding buffer for 30 min at RT then processed as above and probed with anti-PDF. For Pretaporter labeling, brains were fixed as above then blocked for 1 h with 4% BSA in detergent-free PBS. Brains were then incubated with rat anti-Pretaporter (1:250, a generous gift from Yoshinobu Nakanishi, Kanazawa University, Japan)[46] for 2 h at RT in 1% BSA in PBS. Brains were then washed for 20 min in PBS, followed by secondary fixation in 4% PFA + 4% sucrose in PBS for 30 min at RT. After secondary fixation, brains were processed as above and probed with anti-PDF.

**Western blots.** Western blots were performed as follows. Two staged brains per extraction were dissected in PBS with protease inhibitor tablets (Roche Diagnostics, 04693132001). Samples in RIPA buffer (150 μM sodium chloride, 1% Triton X-100, 0.5% sodium deoxycholate, 0.1% sodium dodecyl sulfate, and 50 μM Tris) were immediately flash-frozen on dry ice and stored at −80 °C for <1 week. Samples were thawed on ice and then sonicated for 20 s (Sonifier, Branson, setting 90% duty, output 2), vortexed for 5 s (Standard Mini Vortexer, VMR Scientific Products, speed 4), and centrifuged for 10 min at $11,750 \times g$. Lysate (12 μl) was transferred to tubes with NuPage LDS buffer (4 μl, Invitrogen, NP007) and 2-mercaptoethanol (0.8 μl, Sigma-Aldrich, M7154), then vortexed as above. Samples were incubated at RT for 20 min, then boiled at 100 °C for 10 min, followed by centrifugation at $16,000 \times g$ for 10 min. Equal volumes of lysate were loaded into precast 4–12% Bis–Tris gels (Invitrogen, NP0336) with NuPage MES running buffer (Invitrogen, NP002). NuPage antioxidant (Invitrogen, NP0005) was added to the middle chamber to ensure 2-mercaptoethanol entered gels. Samples were run at constant 100 V for 10 min, then at constant 150 V until dye exited. Samples were then transferred to nitrocellulose membranes (PROTRAN, NBA085C001EA) overnight at 4 °C in NuPage transfer buffer supplemented with 20% methanol with constant 30 mA. Following transfer, membranes were rinsed with deionized water and air-dried for 1 h. Membranes were then blocked with 2% powdered milk in PBS-T (0.1% Tween-20, 150 mM NaCl, 5 mM KCl, and 25 mM Tris, pH 7.6) for 1 h at RT with rotation. Membranes were then incubated in primary antibodies diluted in 2% powdered milk in PBS-T overnight at 4 °C with rotation, then washed 6× for 5 min in PBS-T at RT. Secondary antibodies were then applied in 2% milk in PBS-T for 1 h at RT with rotation, followed by 6 × 5 min PBS-T washes, then imaged using an Odyssey imager (LI-COR Biosciences). Intensity measurements were taken with Li-Cor Image Studio Lite and then normalized to the α-tubulin control. Primary antibodies used: rabbit anti-α-tubulin (Abcam, AB52866, 1:20,000), mouse anti-Draper (DSHB, Draper 8A1, 1:500), goat anti-Ced-12 (1:1500, gift from Erika Geisbrecht, Kansas State University, USA)[57], rat anti-Ced-6 (1:250, gift from Takeshi Awasaki, Kyorin University, Japan)[9], chicken anti-Src42a (1:500, gift from Shigeo Hayashi, Riken, Japan)[56], and rabbit anti-Drk (1:2000, gift from Efthimios Skoulakis, Biomedical Sciences Research Center Alexander Fleming, Greece)[58]. Secondary antibodies used: AlexaFlour 800 goat anti-mouse, AlexaFlour 680 donkey anti-goat, AlexaFlour 680 goat anti-rat, AlexaFlour 790 goat anti-chicken, and AlexaFlour 680 goat anti-rabbit, all at a dilution of 1:10,000. For stripping/reprobing, blots were washed 5 min at RT with deionized water, followed by 5 min in 0.2 N NaOH, then washed again for 5 min with deionized water. Blots were then blocked and probed as above.

**Statistical analyses.** All statistics were performed using Prism software (Graph-Pad, San Diego, CA). All data sets were subject to D'Agostino's normality tests and ROUT outlier tests with Q set to 1%. All groups that met the criteria for normality were analyzed with an unpaired two-tailed student's $t$ test (2 comparison) or with a one-way ANOVA (3 + comparison), followed by Tukey's multiple comparison tests for comparisons between means. For all nonparametric data sets, Mann–Whitney tests (two comparison) or Kruskal–Wallis tests (3 + comparison) followed by Dunn's multiple comparison tests were performed. Different test were used based on normality as directed (GraphPad Prism). Parametric tests, including t-tests and ANOVAs, assume normality while nonparametric tests do not. Therefore, nonparametric tests (Mann–Whitney and Kruskal–Wallis) were used when data did not pass normality standards. For all data sets $n$ = number of animals (imaging data) or n=number of independent protein extractions (Western blots), with scatter plot graphs used in all figures to show every data point. Significance in all figures is designated as follows: $p > 0.05$ (not significant, N.S.), $p < 0.05$ (*), $p < 0.01$ (**), $p < 0.001$ (***) and $p < 0.00001$ (****).

**Reporting summary.** Further information on research design is available in the Nature Research Reporting Summary linked to this article.

## Data availability

All data supporting the findings of this study are provided within the paper and its supplementary information. All additional information and *Drosophila* lines generated for this study will be made available upon reasonable request to the authors. Source data are provided with this paper.

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

## Acknowledgements

We would most like to thank the Bloomington *Drosophila* Stock Center (Indiana University, USA) as well as the Vienna *Drosophila* Resource Center (Vienna Biocenter, Austria) for essential genetic stocks, and the Developmental Studies Hybridoma Bank (DHSB; University of Iowa, USA) for key antibodies. We are especially grateful to Dr. Fen-Biao Gao (University of Massachusetts Medical Center, USA), Dr. Erika Geisbrecht (Kansas State University, USA), Dr. Takeshi Awasaki (Kyorin University, Japan), Dr. Shigeo Hayashi (Riken Institute, Japan), and Dr. Efthimios Skoulakis (Biomedical Sciences Research Center Alexander Fleming, Greece) for kindly providing their critical antibodies. We thank members of the Broadie lab for their constructive input throughout the course of this long study. This work has been entirely supported by NIH grant MH084989 to K.B.

## Author contributions

D.J.V. and K.B. developed the concepts and designed the experiments. D.J.V. conducted the majority of experiments and data analyses. C.J.M. performed the experiments and data analyses in Figs. 2c and 3b–d. D.J.V. and K.B. cowrote the paper. K.B. oversaw the project.

## Competing interests

The authors declare no competing interests.
