## [Peer Review File · Nature Communications]

Reviewers' Comments:

Reviewer #1:

Remarks to the Author:

The manuscript by Vita et al. exposes neuron-glia signaling that mediates removal of the developmentally transient PDF-Tri neurons in the *Drosophila* adult brain. These neurons are normally eliminated during several days after eclosion but remain functional in the *Drosophila* FXS model/*fmrp* null mutants. The authors uncover the role of ensheathing and cortex glia in elimination of the PDF-Tri neurons, which is mediated by the phagocytic transmembrane receptor Draper and Dynamin, and reveal that neuron specific expression of FMRP promotes Draper expression in glia through InR signaling in glial cells.

In general, the study is interesting and uncovers a new pathway of neuron-glia interactions during developmental remodeling. Due to the current world situation I am reluctant to ask for additional experiments but it would be interesting to know whether glia act just as removers of dying neurons or they are actually needed for neuronal death. Detecting persisted neurons with the specific antibody does not answer this question. Dead neurons could be accumulating inside the glial cells because of their impaired degradation (Draper is required for degradation) or they may stay unengulfed and functional as in the *fmrp* null mutant. The important question remains unclear to me, whether PDF-Tri neurons remain functional in the *draper* null or RNAi knockdown flies as in the FXS model?

An interesting result is that *draper* overexpression in glia rescues *fmrp* null phenotype, suggesting that Draper induces neuronal death or elimination by phagocytosis, phagoptosis. Do the authors detect cell death markers in these flies? What does Draper recognize on surfaces of PDF-Tri neurons?

Reduced Shrub levels rescue *fmrp* null phenotype, but there is no evidence that it affects glial phagocytosis. Therefore, two last sentences in "Results" should be removed/clarified. Do InRp levels change in *shrub/+;fmr1/fmr1*? Do Draper levels change in these flies?

Figure 1: there is no value of scale bars.

Figure 2: the brain in panel c looks different from panels a and b, staining is barely seen in optic lobes compared to a and b.

In the text, line 209 there is a mistake in control data: "The glial-specific Draper knockdown causes significantly increased PDF-Tri neuron area (t-test, $p < 0.0001$, 582.24 ± 14.41 $n = 16$ control..."

It is important to show on the picture which driver was used for *shibire* ts and *draper* RNAi (*repo-QF > shibire* ts or *repoGal4 > draper* RNAi).

Figure 3: in control of CG there is very little staining in optic lobes. Is there reason for this?

Figure 5: In WB analyses normalization should be done with Actin or Tubulin and not with total protein. Full blots should be presented to see marker and other bands. *draper* null and overexpression samples will help to recognize the specific bands for Draper I and Draper II/III.

Figure 7: Merge picture of glial labeling and InRp is missing. It is hard to see the overlap between InRp and glial membranes. The picture of InRCA overexpression is missing too. What are the colors?

Figure 8: There is limited colocalization of Shrub and PDF in panel 1. Do InRp levels change in the *shrub* mutant/heterozygous?

English editing is needed. There are numerous spelling and grammar mistakes that must be fixed. Description of statistical analysis is not clear. It is not consistent in different figures. Sometimes P value is given in numbers, sometimes in $< >$, different tests are performed in different experiments and it is hard to understand the reason why.

Genotypes are sometimes in italics and sometimes are not. Gal4 or gal4, N.S. or non significant. Must be consistent in the text and figure legends.

The fluorescent pictures are in different color schemes. It should be consistent.

Line 43: "...to initiate glial engulfment phagocytosis..." What does it mean? Engulfment is one of steps in phagocytosis.

Reviewer #2:

Remarks to the Author:

The Vita et al. manuscript details a neuronal function for FMRP in promoting the elimination of a subset of peptidergic neurons in the *Drosophila* brain. The authors demonstrate that neuronal FMRP is required for the clearance of PDF-Tri neurons by two populations of fly glia. They provide convincing data that neuronal FMRP drives Insulin receptor activation in glia which in turn activates the key engulfment receptor Draper. On the neuronal side, they demonstrate that the FMRP target Shrub dominantly suppresses the clearance defect displayed by *dfmr1* mutants, suggesting that elevated levels of Shrub in *dfmr1* mutants impair clearance of PDF-Tri neurons. These findings are interesting as they contribute to the growing consensus that the connectivity deficits observed in FXS models are not solely the result of dysregulated cell-autonomous pathways, but also reflect disrupted neuron-glia interactions. There is added novelty in the discovery that InR signaling may drive *Drpr* expression during normal development, and not just in an injury context as previously reported (Musashe et al., 2016). However, the data in this paper do not hang together in a coherent story. They are not internally consistent, and are not easily consistent with previous studies from either the PI's or other labs.

1. In Gatto and Broadie (2011), the PI's group demonstrated that FMRP is required not only for clearance of the PDF-Tri neurons, but also for the decision of these cells to die in the first place, since TUNEL labeling is greatly reduced in *dfmr1* LOF. These data argue that *dfmr1* mutants have defective programmed cell death and that the mutant phenotype does not reflect solely a defect in glial phagocytosis or fragmentation. It is possible that these data hint at a function for phagoptosis, or glia-mediated killing, of PDF-Tri neurons, but this is not discussed. It is not clear how to reconcile these disparate findings.

2. In Musashe et al. (2016), the authors find that baseline InR signaling does not influence glial *Drpr* at either the transcriptional or protein levels. They argue that InR signaling acts via a STAT92E-dependent transcriptional program to upregulate *Drpr* only in situations of high phagocytic demand, such as following injury. This discrepancy should be discussed. Are the authors proposing a general requirement for InR signaling in regulating baseline *Drpr*? Or is this requirement specific to the PDF-Tri neurons?

3. In O'Connor et al. (2017), the authors find that FMRP is required in glia to promote clearance of pruned mushroom body axons (in the absence of a change in baseline *Drpr*). Given these data, it is perhaps surprising that the current study finds only a neuronal requirement for FMRP in "clearance" of this peptidergic cell type. Again, since the authors already published that PDF-Tri neurons do not undergo programmed cell death in *dfmr1* mutants, it is unclear whether the primary defect here is one of glial clearance.

4. The explanation of the Shrub findings are not logical. The authors frame the ESCRT-III component Shrub as a pro-clearance protein. They claim that it is active during neuron clearance to physically fragment the neuron or perhaps to facilitate glial phagocytosis. In the abstract, they state that FMRP activates Shrub to drive PDF-Tri neuron clearance. But this is the opposite of what their data show, and is also in opposition to their previous published work demonstrating that FMRP is a negative regulator of Shrub. In Figure 8, they demonstrate that *shrub* dominantly suppresses the persistent PDF-Tri neuron phenotype observed in *dfmr1* mutants. In other words, reducing the dosage of *shrub* in a *dfmr1* background improves clearance. This is the genetic result that would be expected for an inhibitor, not an activator, of clearance/engulfment. It is totally unclear how to explain their findings with respect to Shrub, or how this fits in to the InR-*Drpr* link presented earlier in the paper.

Reviewer #3:

Remarks to the Author:

Vita and colleagues use a developmental pruning model in newly eclosed *Drosophila* (PDF-Tri neurons) to explore the role of glial phagocytic removal of pruned cells and projections and contributions of FMRP. The authors show that clearance of PDF-Tri neurons is delayed in FMRP mutants (model for Fragile X), and their genetic analyses suggest that this phenotype results from deficient activation of the glial Insulin Receptor and Draper pathways, which are known to drive glial engulfment. The authors also explore the role for the ESCRT-III molecule (Shrub) in PDF-Tri pruning and suggest that overexpression of Shrub within FMRP-deficient neurons contributes to delayed fragmentation and, thus, glial clearance. Overall, this work offers a useful model to explore the connections between FMRP function and the non-autonomous effects on glial engulfment pathways. The experiments are properly controlled and statistical analysis and data presentation is appropriate. The mechanistic role for Shrub is, as presented, a bit unclear/weak. The following points below should be addressed to strengthen the model and the manuscript.

Major points:

1. Results in Figure 3 do implicate both ensheathing and cortex glia in clearance of PDF-Tri neurons. However, the authors suggest that they function cooperatively and in spatially distinct manners (see lines 242-246). Clearance phenotypes may be stronger with repo-Gal4 due the strength of the promoter. Draper immunostainings may be one way to demonstrate the efficacy of each driver (repo versus EG, CG).

2. In addition to the above point, the authors suggest that each glial subtype functions within "their two spatial domains" (line 245-246). This idea is not unprecedented based on previously published work that different glial subtypes clear neuronal cell bodies versus projections. However, the results here do not support that statement. The authors could use PDF-Gal4 to express membrane GFP and a nuclear marker (NLS tagged molecule) and compare clearance in Draper RNAi animals to determine if CG are responsible for cell body clearance while EG are required for proper phagocytosis of projections.

3. Figure 7 could be improved to strengthen the authors' claims. For example, InR-P in the glial cell bodies is highlighted with arrows, but there is no marker to denote glial nuclei (repo stain or NLS-tagged marker). Repeating these experiments in GFP-tagged EG and CG would be relatively straight forward and support the model that both glial cells types are promoting clearance in a InR/Draper manner.

In addition, to further support this model genetically, the authors should show that loss of InR in glia results in delayed clearance of PDF-Tri neurons (with InR RNAi, etc.)

4. The final Figure (8) and the role of Shrub in this model is unclear. The authors state that FMRP inhibits Shrub translation/expression. However, Shrub is a positive regulator of fragmentation. Why do high levels of Shrub perturb normal fragmentation of neurons? One would expect faster (or at least not delayed) fragmentation of projections with higher Shrub levels. In addition, the first part of Figure 8 claims to show accumulation of Shrub at fragmentation sites, but the marker for neurons is PDF (not a membrane marker). The authors should show Shrub localization in cells that express a membrane marker. Finally, some genetic approaches would strengthen this model. For example: Does forced overexpression of Shrub in PDF-Tri neurons delay fragmentation? Reduced InR-P? Reduced Draper?

Minor points:

1. It would be useful to state the label shown within each Figure. For example, anti-PDF in Figure 2, etc.

2. Check spelling/grammar errors throughout (lines 276, 382, 397 caught my eye).

We submit here our revised NCOMMS-20-18469 for your specified revision deadline. Given the passage of time during this incredibly challenging SARS-CoV-2 pandemic, please remember your gracious revision extension for this study (your Sept 17 email). Below, we provide our point-by-point responses to all three Reviewer full comments. This revision contains 11 entirely new figures, including 2 main text figures (Figures 4 and 8) and 9 new supplemental figures. In response to Reviewer suggestions, there are also numerous modifications of the original 8 figures, as detailed in our point-by-point responses below. This revision now includes 19 total figures; 10 main + 9 supplemental. All new text in the revised manuscript based on Reviewer comments is marked in blue. We have carefully adhered to Journal guidelines, so we believe the attached manuscript meets all requirements for publication. Please let us know if anything further is needed.

Reviewer #1 (Remarks to the Author):

The manuscript by Vita et al. exposes neuron-glia signaling that mediates removal of the developmentally transient PDF-Tri neurons in the *Drosophila* adult brain. These neurons are normally eliminated during several days after eclosion but remain functional in the *Drosophila* FXS model/*fmrp* null mutants. The authors uncover the role of ensheathing and cortex glia in elimination of the PDF-Tri neurons, which is mediated by the phagocytic transmembrane receptor Draper and Dynamin, and reveal that neuron specific expression of FMRP promotes Draper expression in glia through InR signaling in glial cells. In general, the study is interesting and uncovers a new pathway of neuron-glia interactions during developmental remodeling. Due to the current world situation I am reluctant to ask for additional experiments but it would be interesting to know whether glia act just as removers of dying neurons or they are actually needed for neuronal death. Detecting persisted neurons with the specific antibody does not answer this question. Dead neurons could be accumulating inside the glial cells because of their impaired degradation (Draper is required for degradation) or they may stay unengulfed and functional as in the *fmrp* null mutant. The important question remains unclear to me, whether PDF-Tri neurons remain functional in the *draper* null or RNAi knockdown flies as in the FXS model?

As suggested, we tested neuronal death in *draper* mutants using TUNEL DNA labeling (new Supplemental Figure 1). PDF-Tri neurons die in null mutants and the unengulfed TUNEL-positive neurons persist. Glial Draper function is required for their clearance. Since PDF-Tri neurons are dead, they do not remain functional in *draper* null mutants.

An interesting result is that *draper* overexpression in glia rescues *fmrp* null phenotype, suggesting that Draper induces neuronal death or elimination by phagocytosis, phagoptosis. Do the authors detect cell death markers in these flies? What does Draper recognize on surfaces of PDF-Tri neurons?

As suggested, we tested for candidate cell death markers and Draper surface cues (new Supplemental Figure 2). Specifically, we assayed PDF-Tri neurons for Pretaporter and phosphatidylserine (Annexin V). We find no evidence of their surface expression. We consider these the best-established candidates, although others will be tested later.

Reduced Shrub levels rescue *fmrp* null phenotype, but there is no evidence that it affects glial phagocytosis. Therefore, two last sentences in "Results" should be removed/clarified.

We have clarified these two sentences in the revised Results section.

Do *InRp* levels change in *shrub/+;fmr1/fmr1*? Do Draper levels change in these flies?

As suggested, we tested both *InRp* and Draper levels in *shrub/+; dfmr1/dfmr1* mutants (new Supplemental Figures 8, 9). We find reduced Shrub level in *dfmr1* nulls mitigates *InRp* and Draper phenotypes, similar to effects on PDF-Tri neuron clearance defects.

Figure 1: there is no value of scale bars.

We apologize for the oversight. Scale bar values have been added to the figure legend.

Figure 2: the brain in panel c looks different from panels a and b, staining is barely seen in optic lobes compared to a and b.

In response to this query, we have replaced the Figure 2 panel c image with another image example that more clearly shows optic lobe staining (revised Figure 2).

In the text, line 209 there is a mistake in control data: "The glial-specific Draper knockdown causes significantly increased PDF-Tri neuron area (t-test, $p < 0.0001$, 582.24 ± 14.41 n=16 control..."

We have corrected this text mistake.

It is important to show on the picture which driver was used for *shibire ts* and draper RNAi (*repo-QF>shibire ts* or *repoGal4>draperRNAi*).

We have indicated the drivers directly on the figure.

Figure 3: in control of CG there is very little staining in optic lobes. Is there reason for this?

In response to this query, we have replaced the Figure 3 CG control image with another image example that more clearly shows optic lobe staining (revised Figure 3).

Figure 5: In WB analyses normalization should be done with Actin or Tubulin and not with total protein. Full blots should be presented to see marker and other bands. *draper* null and overexpression samples will help to recognize the specific bands for Draper I and Draper II/III.

As suggested, we have added Western blots normalized to alpha tubulin, and revised

the quantification in Figure 5 (revised Figure 6). We have also included the full Western blots in the supplementary material (new Supplemental Figures 4 and 5).

Figure 7: Merge picture of glial labeling and InRp is missing. It is hard to see the overlap between InRp and glial membranes. The picture of InRCA overexpression is missing too. What are the colors?

As suggested, we added a merged InRp/GFP image to Figure 7 (revised Figure 9), with the heat-map scale to illustrate the color scheme of the InRp labeling intensity. We also added InRp labeling in PDF-Tri associated glia subtypes (new Supplemental Figure 6). There is no image of InRCA overexpression as the only antibody available is anti-InRp.

Figure 8: There is limited colocalization of Shrub and PDF in panel 1. Do InRp levels change in the shrub mutant/heterozygous?

We have included Shrub labeling in *PDF-Gal4>GFP* marked PDT-Tri neurons to further confirm the Shrub localization at fragmentation foci (new Supplemental Figure 7). As suggested, we have also now tested InRp levels in the *shrub/+; dfmr1/dfmr1* mutants (new Supplemental Figure 8) to show a mitigated InRp defect (see new Discussion).

English editing is needed. There are numerous spelling and grammar mistakes that must be fixed.

We apologize for the mistakes, and we have done our best to correct them.

Description of statistical analysis is not clear. It is not consistent in different figures. Sometimes P value is given in numbers, sometimes in < >, different tests are performed in different experiments and it is hard to understand the reason why.

We have clarified why we use multiple statistical tests (*Statistical Analyses Methods*). The statistical package used for all the analyses in this entire study, GraphPad Prism, calculates P-values to 4 decimal places. More significant changes are reported by the program simply as <. This is why small P-values are exact, while large P-values are not.

Genotypes are sometimes in italics and sometimes are not. Gal4 or gal4, N.S. or non significant. Must be consistent in the text and figure legends.

We have corrected issues with text consistency. Genotypes are italicized for an allele (*draper*) or insertion (*draper-RNAi*). Proteins are not italicized, but capitalized (Draper). We have also tried to be consistent in the use of "Gal4" and "N.S." throughout the text.

The fluorescent pictures are in different color schemes. It should be consistent.

Line 43: "...to initiate glial engulfment phagocytosis..." What does it mean? Engulfment is one of steps in phagocytosis.

We have changed this to "We show that FMRP is required in neurons, not glia, for glial

engulfment, indicating FMRP-dependent neuron-to-glia signaling mediates clearance.”

Reviewer #2 (Remarks to the Author):

The Vita et al. manuscript details a neuronal function for FMRP in promoting the elimination of a subset of peptidergic neurons in the *Drosophila* brain. The authors demonstrate that neuronal FMRP is required for the clearance of PDF-Tri neurons by two populations of fly glia. They provide convincing data that neuronal FMRP drives Insulin receptor activation in glia which in turn activates the key engulfment receptor Draper. On the neuronal side, they demonstrate that the FMRP target Shrub dominantly suppresses the clearance defect displayed by *dfmr1* mutants, suggesting that elevated levels of Shrub in *dfmr1* mutants impair clearance of PDF-Tri neurons. These findings are interesting as they contribute to the growing consensus that the connectivity deficits observed in FXS models are not solely the result of dysregulated cell-autonomous pathways, but also reflect disrupted neuron-glia interactions. There is added novelty in the discovery that InR signaling may drive *Drpr* expression during normal development, and not just in an injury context as previously reported (Musashe et al., 2016). However, the data in this paper do not hang together in a coherent story. They are not internally consistent, and are not easily consistent with previous studies from either the PI's or other labs.

1. In Gatto and Broadie (2011), the PI's group demonstrated that FMRP is required not only for clearance of the PDF-Tri neurons, but also for the decision of these cells to die in the first place, since TUNEL labeling is greatly reduced in *dfmr1* LOF. These data argue that *dfmr1* mutants have defective programmed cell death and that the mutant phenotype does not reflect solely a defect in glial phagocytosis or fragmentation. It is possible that these data hint at a function for phagoptosis, or glia-mediated killing, of PDF-Tri neurons, but this is not discussed. It is not clear how to reconcile these disparate findings.

As shown in response to Reviewer 1, point 1, TUNEL labeling reveals PDF-Tri neuronal death in the *draper* mutants (new Supplemental Figure 1). The PDF-Tri neurons die and persist as unengulfed corpses. As the Reviewer suggests, this could be consistent with a phagoptosis mechanism. However, we do not want to over-interpret these revision data, and would not like to make this suggestion without extensive further verification.

2. In Musashe et al. (2016), the authors find that baseline InR signaling does not influence glial *Drpr* at either the transcriptional or protein levels. They argue that InR signaling acts via a STAT92E-dependent transcriptional program to upregulate *Drpr* only in situations of high phagocytic demand, such as following injury. This discrepancy should be discussed. Are the authors proposing a general requirement for InR signaling in regulating baseline *Drpr*? Or is this requirement specific to the PDF-Tri neurons?

As suggested, we further discuss our results in comparison to Musashe et al. 2016. The previous study suggests InR signaling happens under “high phagocytic demand”, and this may appear at odds to the PDF-Tri neuron situation. However, there is similar “high phagocytic demand” in the early post-eclosion brain (Gatto and Broadie, 2011),

and thus our experiments on InR signaling Draper induction are likely not investigating baseline conditions. We have done nothing in this study to test a general requirement for InR signaling in regulating baseline Draper, as all of our work here is focused on the developmental elimination of the PDF-Tri neurons. Therefore, we cannot know whether the requirement is specific to PDF-Tri neurons. We suggest that similarly to axotomy, InR signaling is the means to facilitate Draper dependent removal of PDF-Tri neurons.

3. In O'Connor et al. (2017), the authors find that FMRP is required in glia to promote clearance of pruned mushroom body axons (in the absence of a change in baseline Drpr). Given these data, it is perhaps surprising that the current study finds only a neuronal requirement for FMRP in “clearance” of this peptidergic cell type. Again, since the authors already published that PDF-Tri neurons do not undergo programmed cell death in *dfmr1* mutants, it is unclear whether the primary defect here is one of glial clearance.

We greatly appreciate the thorough cross-referencing of previous studies with our work! However, O'Connor et al. (2017) does not report that FMRP is required within glia to promote the clearance of pruned mushroom body axons. That study used only global *dfmr1* null mutants in the brain, with cell-specific *dfmr1* knockdown employed only for the circulating haemocytes. Therefore, there is no inconsistency with our current study.

4. The explanation of the Shrub findings are not logical. The authors frame the ESCRT-III component Shrub as a pro-clearance protein. They claim that it is active during neuron clearance to physically fragment the neuron or perhaps to facilitate glial phagocytosis. In the abstract, they state that FMRP activates Shrub to drive PDF-Tri neuron clearance. But this is the opposite of what their data show, and is also in opposition to their previous published work demonstrating that FMRP is a negative regulator of Shrub. In Figure 8, they demonstrate that *shrub* dominantly suppresses the persistent PDF-Tri neuron phenotype observed in *dfmr1* mutants. In other words, reducing the dosage of *shrub* in a *dfmr1* background improves clearance. This is the genetic result that would be expected for an inhibitor, not an activator, of clearance/engulfment. It is totally unclear how to explain their findings with respect to Shrub, or how this fits in to the InR-Drpr link presented earlier in the paper.

In our submission, we failed to adequately stress that both Shrub loss-of-function (LOF) and gain-of-function (GOF) cause similar phenotypes, as shown in a number of studies (Babst et al, 2002a; Teis et al., 2008) and also confirmed by us in the *Drosophila* brain (Vita and Broadie, 2017). To further test this well-established conclusion for the PDF-Tri neuron clearance mechanism, we have now targeted Shrub overexpression specifically in these neurons to show impaired glial clearance (new Supplemental Figure 7). We conclude that Shrub LOF/GOF phenocopy, providing an explanation for the apparent discrepancy. We have revised both the abstract and main text to clarify these findings.

Reviewer #3 (Remarks to the Author):

Vita and colleagues use a developmental pruning model in newly eclosed *Drosophila* (PDF-Tri neurons) to explore the role of glial phagocytic removal of pruned cells and

projections and contributions of FMRP. The authors show that clearance of PDF-Tri neurons is delayed in FMRP mutants (model for Fragile X), and their genetic analyses suggest that this phenotype results from deficient activation of the glial Insulin Receptor and Draper pathways, which are known to drive glial engulfment. The authors also explore the role for the ESCRT-III molecule (Shrub) in PDF-Tri pruning and suggest that overexpression of Shrub within FMRP-deficient neurons contributes to delayed fragmentation and, thus, glial clearance. Overall, this work offers a useful model to explore the connections between FMRP function and the non-autonomous effects on glial engulfment pathways. The experiments are properly controlled and statistical analysis and data presentation is appropriate. The mechanistic role for Shrub is, as presented, a bit unclear/weak. The following points below should be addressed to strengthen the model and the manuscript.

Major points:

1. Results in Figure 3 do implicate both ensheathing and cortex glia in clearance of PDF-Tri neurons. However, the authors suggest that they function cooperatively and in spatially distinct manners (see lines 242-246). Clearance phenotypes may be stronger with repo-Gal4 due the strength of the promoter. Draper immunostainings may be one way to demonstrate the efficacy of each driver (repo versus EG, CG).

We appreciate the suggestion and recognize the possible quandary. However, we were unable to successfully answer this question, despite our repeated experimental trials to do so. Each driver expresses in a different number of glia cells in different brain regions. It was not possible to fairly cross-compare Gal4 driver strengths, and we do not wish to unintentionally misinterpret results. We do thoroughly test the hypothesis of cooperative glial function in response to your comment 2 (below) to better address this key question.

2. In addition to the above point, the authors suggest that each glial subtype functions within “their two spatial domains” (line 245-246). This idea is not unprecedented based on previously published work that different glial subtypes clear neuronal cell bodies versus projections. However, the results here do not support that statement. The authors could use PDF-Gal4 to express membrane GFP and a nuclear marker (NLS tagged molecule) and compare clearance in Draper RNAi animals to determine if CG are responsible for cell body clearance while EG are required for proper phagocytosis of projections.

As suggested, we used CG- and EG-Gal4 lines to drive *draper*-RNAi to test each glial class contribution to the PDF-Tri neuron clearance (new Main Figure 4), with additional quantification in the supplemental material (new Supplemental Figure 3). We find that CG>RNAi causes the selective retention of PDF-Tri neuron cell bodies, while EG>RNAi has no impact on their removal. Furthermore, we examined the CG/EG contributions to PDF-Tri neuron clearance in both distal and proximal regions. We find CG>RNAi results in clearance defects specifically in proximal regions. EG>RNAi shows less difference between proximal and distal processes. This work supports the conclusion that different glial subtypes clear cell bodies versus projections (e.g. Tasdemir-Yilmaz and Freeman, 2014), and that the two glial classes likely worked together to remove PDF-Tri neurons.

3. Figure 7 could be improved to strengthen the authors' claims. For example, InR-P in the glial cell bodies is highlighted with arrows, but there is no marker to denote glial nuclei (repo stain or NLS-tagged marker). Repeating these experiments in GFP-tagged EG and CG would be relatively straight forward and support the model that both glial cells types are promoting clearance in a InR/Draper manner.

In addition, to further support this model genetically, the authors should show that loss of InR in glia results in delayed clearance of PDF-Tri neurons (with InR RNAi, etc.)

As suggested, we repeated InRp imaging with both anti-Repo labeling and GFP-tagged EG/CG classes (new Supplemental Figure 6). Glia co-label for both InRp and Repo, and EG>GFP shows clear co-localization with InRp. CG>GFP also localizes with InRp, but InRp is present broadly within the cortex, making interpretation more difficult as reported previously (Musashe et al. 2016). We also show that loss of InR signaling impairs the PDF-Tri neuron glial clearance (new Main Figure 8), although glial-targeted InR-RNAi alone proved insufficient to block the glial clearance mechanism due to limitations with the available InR-RNAi lines, as has been reported previously (Mushashe et al., 2016).

4. The final Figure (8) and the role of Shrub in this model is unclear. The authors state that FMRP inhibits Shrub translation/expression. However, Shrub is a positive regulator of fragmentation. Why do high levels of Shrub perturb normal fragmentation of neurons? One would expect faster (or at least not delayed) fragmentation of projections with higher Shrub levels. In addition, the first part of Figure 8 claims to show accumulation of Shrub at fragmentation sites, but the marker for neurons is PDF (not a membrane marker). The authors should show Shrub localization in cells that express a membrane marker. Finally, some genetic approaches would strengthen this model. For example: Does forced overexpression of Shrub in PDF-Tri neurons delay fragmentation? Reduced InR-P? Reduced Draper?

As discussed above (Reviewer 2, point 4), Shrub LOF and GOF phenocopy each other (Babst et al, 2002a; Teis et al., 2008; Vita and Broadie, 2017). As suggested, we now test the Shrub localization within PDF-Tri neurons labeled with membrane-bound GFP (new Supplemental Figure 7a). We find Shrub puncta at areas of narrowed PDF-Tri neuron processes, consistent with a role in fragmentation. As suggested, we now show Shrub overexpression in PDF-Tri neurons impairs their clearance by glial phagocytosis (new Supplemental Figure 7b,c). We hope that this work helps clarify the role of Shrub.

Minor points:

1. It would be useful to state the label shown within each Figure. For example, anti-PDF in Figure 2, etc.

All figures now state the label directly on the figure (e.g. anti-PDF in Figure 2).

2. Check spelling/grammar errors throughout (lines 276, 382, 397 caught eye).

We apologize for these mistakes, and we have done our best to correct them.

Reviewers' Comments:

Reviewer #1:

Remarks to the Author:

The authors substantially revised the manuscript and addressed most of my concerns.

Reviewer #2:

Remarks to the Author:

This manuscript has been improved from the initial version. Most notably, the directionality of the FMRP-Shrub regulatory relationship has been fixed. As a minor point, GMR54H02Gal4 is not specific for cortex glia, but is also expressed in cortex glia (Countinho-Budd et al., 2017).

In my opinion, this paper still does not represent an important conceptual advance suitable for Nature Comm.

Reviewer #3:

Remarks to the Author:

The authors have addressed my concerns in the revised version of this manuscript. One (minor) suggestion is to provide a higher magnification/additional inset to Supplemental Fig 6, panel C to show InR-P and EG labeling more convincingly.

Reviewer #1:

The authors substantially revised the manuscript and addressed most of my concerns.

We thank the Reviewer.

Reviewer #2:

This manuscript has been improved from the initial version. Most notably, the directionality of the FMRP-Shrub regulatory relationship has been fixed. As a minor point, GMR54H02Gal4 is not specific for cortex glia, but is also expressed in cortex glia (Countinho-Budd et al., 2017).

In my opinion, this paper still does not represent an important conceptual advance suitable for Nature Comm.

We thank the Reviewer, and agree GMR54H02Gal4 is expressed in cortex glia.

Reviewer #3:

The authors have addressed my concerns in the revised version of this manuscript. One (minor) suggestion is to provide a higher magnification/additional inset to Supplemental Fig 6, panel C to show InR-P and EG labeling more convincingly.

We thank the Reviewer, and as suggested we include an increased magnification inset in the merge of supplemental Figure 6c.